# Yeast cell fate control by temporal redundancy modulation of transcription factor paralogs

Yan Wu [1,2,3,4], Jiaqi Wu[1,2,3], Minghua Deng[1,4,5] & Yihan Lin [1,2,3✉]

Recent single-cell studies have revealed that yeast stress response involves transcription factors that are activated in pulses. However, it remains unclear whether and how these dynamic transcription factors temporally interact to regulate stress survival. Here we show that budding yeast cells can exploit the temporal relationship between paralogous general stress regulators, Msn2 and Msn4, during stress response. We find that individual pulses of Msn2 and Msn4 are largely redundant, and cells can enhance the expression of their shared targets by increasing their temporal divergence. Thus, functional redundancy between these two paralogs is modulated in a dynamic manner to confer fitness advantages for yeast cells, which might feed back to promote the preservation of their redundancy. This evolutionary implication is supported by evidence from Msn2/Msn4 orthologs and analyses of other transcription factor paralogs. Together, we show a cell fate control mechanism through temporal redundancy modulation in yeast, which may represent an evolutionarily important strategy for maintaining functional redundancy between gene duplicates.

[1] Center for Quantitative Biology, Academy for Advanced Interdisciplinary Studies, Peking University, Beijing, China. [2] The MOE Key Laboratory of Cell Proliferation and Differentiation, School of Life Sciences, Peking University, Beijing, China. [3] Peking-Tsinghua Center for Life Sciences, Academy for Advanced Interdisciplinary Studies, Peking University, Beijing, China. [4] School of Mathematical Sciences, Peking University, Beijing, China. [5] Center for Statistical Science, Peking University, Beijing, China. ✉email: yihan.lin@pku.edu.cn

Cells need to make decisions in responding to changing environment[1]. Cellular decision-making involves two successive steps: the encoding of environmental inputs into intracellular signals, followed by the decoding of intracellular signals into corresponding downstream decisions[2]. Recent single-cell studies have found that kinases and transcription factors (TFs) in signal encoding circuits are often activated in a highly dynamic fashion[2–5]. These dynamics can be stochastic (i.e., affected by noise), repetitive, and often appear as stereotyped pulses[5,6]. At the level of individual cells, the characteristics of such dynamics are often modulated by environmental inputs, which are then decoded to affect downstream physiology[4,7]. At the population level, such dynamics could result in heterogeneous cell states, enabling bet-hedging strategies for coping with fluctuating environments[8].

Pulsatile regulators have been shown to play key roles in regulating various physiological processes, ranging from microbial stress responses[6,9–17] to animal cell signaling[7,13,18–24]. Because some of these processes are simultaneously regulated by multiple pulsatile regulators[14,16,21,25–27], establishing the regulatory principles for these processes requires understanding how different dynamic regulators work together to encode upstream inputs and to control downstream cell fates[5]. To do so, one would need to characterize the temporal interactions between different regulators in individual cells, and decipher whether and how such temporal interactions lead to downstream physiological effects in the same cell. However, this is technically challenging and remains to be explored in microbial systems.

*Saccharomyces cerevisiae* (i.e., budding yeast) has been utilized as a model organism for studying the mechanism and function of pulsatile TFs[6,9,12,16,25,26,28–31]. A key reason is that several stress-responsive yeast TFs exhibit sustained pulsatile dynamics after the initial response to a step input[6,14,29]. More specifically, the immediate response to a step input is typically synchronized across the cell population (which we term the transient phase of the response), while the second phase of the response is typically stochastic and unsynchronized among cells, constituting a steady-state response at the population level (i.e., the averaged response across cells is steady and largely unchanged). While the transient response dynamics can encode information about the inputs[26], the sustained dynamics after the transient phase enable complex signal processing mechanisms[14,16] and likely play a key role in determining stress response outcome.

Msn2 and Msn4 are general stress response regulators in budding yeast. They both exhibit nuclear-cytoplasmic shuttling dynamics (i.e., dynamic pulses) at the timescale of minutes during stress responses[28], and bind to stress response elements to co-regulate >200 genes[32]. More importantly, Msn2 and Msn4 are paralogous TFs that have been maintained for ~100 million years after the whole-genome duplication (WGD) event[33,34]. Recent studies have investigated their redundant and divergent roles in target activation[31,35], and an intriguing mechanism based on *cis*-regulatory divergence was proposed to account for the maintenance of functional redundancy between this pair of paralogs[35]. Yet, it remains largely unclear whether cells could exploit the temporal interactions between these two paralogs to gain fitness advantages, and whether such advantages might be evolutionarily relevant.

In this work, by integrating single-cell imaging, transcriptome analysis, and mathematical modeling, we show that budding yeast cells can modulate temporal redundancy between Msn2 and Msn4 to enhance stress survival. Mechanistically, the temporal relationship between redundant Msn2 and Msn4 pulses is modulated by altering upstream phospho-regulation to increase the dynamic range of stress response genes. Furthermore, the fitness advantage conferred by temporal redundancy modulation

could help to retain functional redundancy between the paralogs, implicating a potential role of temporal redundancy modulation for the maintenance of redundant paralogs at the evolutionary timescale. Additional experimental characterizations of Msn2/Msn4 orthologs in a related yeast species *Candida glabrata*, as well as analyses on large-scale yeast and human datasets, provide support for this potential role. Together, our work identifies temporal redundancy modulation as a mechanism for yeast cell fate control, and suggests an intriguing role of this mechanism in evolution.

## Results

**Input modulates the temporal relationship between Msn2 and Msn4 pulses.** While the dynamics of Msn2 and Msn4 display relatively high temporal correlations in individual cells[31], previous data indicated the presence of divergent dynamics between these two TFs[25,35]. Yet, it is unclear whether such divergence arises from stochasticity in the dynamics or from regulation by the environmental signals. We thus set out to systematically analyze the divergent dynamics between Msn2 and Msn4, and address how such divergence is regulated by different inputs. More specifically, while both Msn2/Msn4 are activated in pulses in response to diverse stresses, we needed to decipher whether their dynamics are differentially modulated in different cells and under different stress conditions.

To do so, we used a yeast strain carrying endogenously tagged Msn2-CFP and Msn4-YFP, and monitored the dynamic responses of Msn2/Msn4 to various stresses at the single-cell level using a microfluidic device[36] (Fig. 1a, b, Supplementary Fig. 1, and Supplementary Movie 1; see strain list in Supplementary Table 1). In addition to computing cross-correlation functions between their dynamics, we analyzed the temporal relationship between Msn2 and Msn4 dynamics at the level of individual pulses and calculated the fraction of pulses that coincide (i.e., pulse coincidence rate) during steady-state responses. We found that pulses of Msn2/Msn4 are stochastic (Supplementary Fig. 2), and display a relatively high degree of coincidence under low stress levels (Fig. 1c and Supplementary Fig. 3a, b). As the stress level increases, the rate of pulse coincidence decreases, leading to largely divergent pulsatile dynamics for the paralogs. Similarly, cross-correlation between Msn2 and Msn4 traces at time lag zero also decreases as stress level increases, consistent with the trend in pulse coincidence rate (Supplementary Fig. 2c, d). By examining single-cell data, we found that pulse coincidence rate is bimodally distributed under the same stress condition (Fig. 1d and Supplementary Fig. 3a), and the distribution is stress level-dependent (Supplementary Fig. 3c).

We performed further analysis to ensure that the bimodality, as well as the stress-dependent modulation, is not due to differences in pulse frequency (Supplementary Figs. 1b, c and 3d). More specifically, we first withdrew a subpopulation of cells from each of the different conditions to ensure that these subpopulations have comparable pulse frequencies, and then used these subpopulations to compute pulse coincidence rate. Together, these results demonstrated that yeast cells modulate the temporal relationship between Msn2 and Msn4 in response to different stresses, and such temporal relationship displays cell-to-cell variability, resembling the bet-hedging strategy commonly implemented by microbial cell populations to combat stresses[37].

**Individual pulses of Msn2 and Msn4 display functional redundancy.** To investigate how cells might exploit the temporal relationship between Msn2/Msn4 dynamics, we needed to first determine whether pulses of Msn2 and Msn4 are functionally

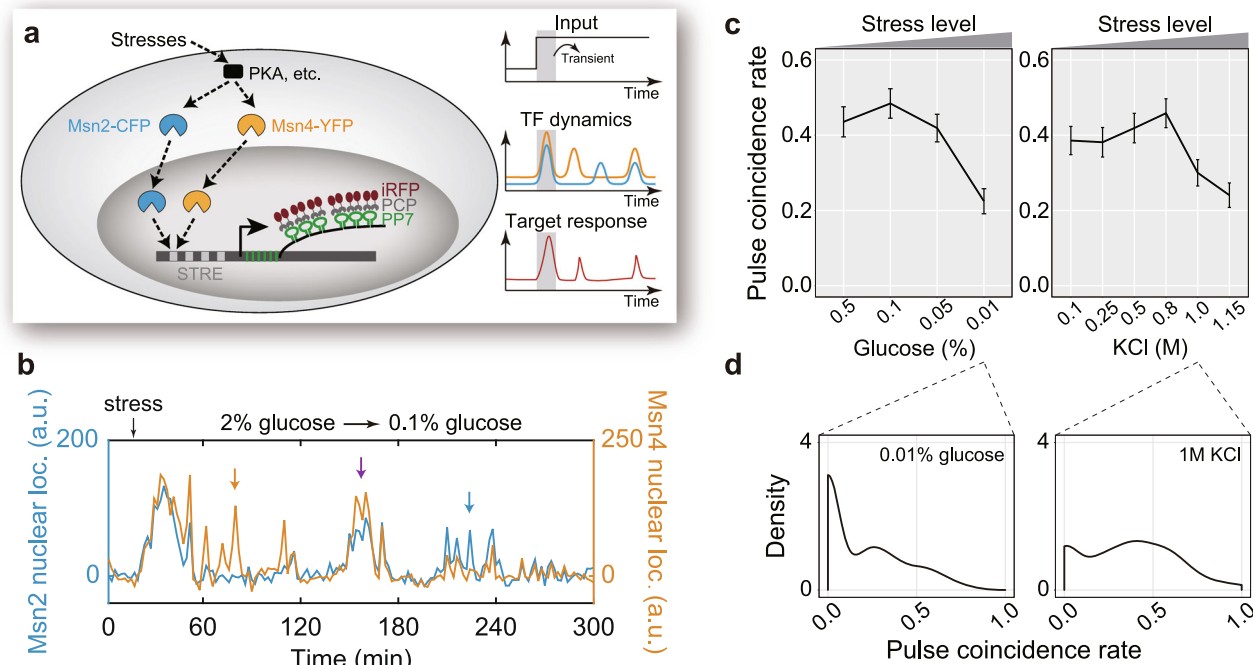

**Fig. 1 Input modulates the temporal relationship between Msn2 and Msn4 pulses. a** Schematics of the single-cell assay for monitoring Msn2/Msn4 dynamics (with fluorescent protein fusions) and target responses (with the PP7-PCP system) in the same cell. Using microfluidics, both transient (shaded) and steady-state responses were recorded. **b** Example single-cell traces of Msn2 (blue) and Msn4 (orange) nuclear localization dynamics in response to a step change in glucose concentration. Coincident pulse (purple arrow) and noncoincident pulses (orange or blue arrow) are indicated. a.u., arbitrary unit. See also Supplementary Movie 1. **c** Pulse coincidence rates between Msn2 and Msn4 under different stress conditions. Cell numbers are 139, 176, 333, and 282 for glucose limitation stresses and 148, 182, 207, 181, 195, and 201 for osmotic stresses. Error bars indicate 95% confidence intervals (CIs) and centers indicate means. **d** Distributions of pulse coincidence rates in individual cells.

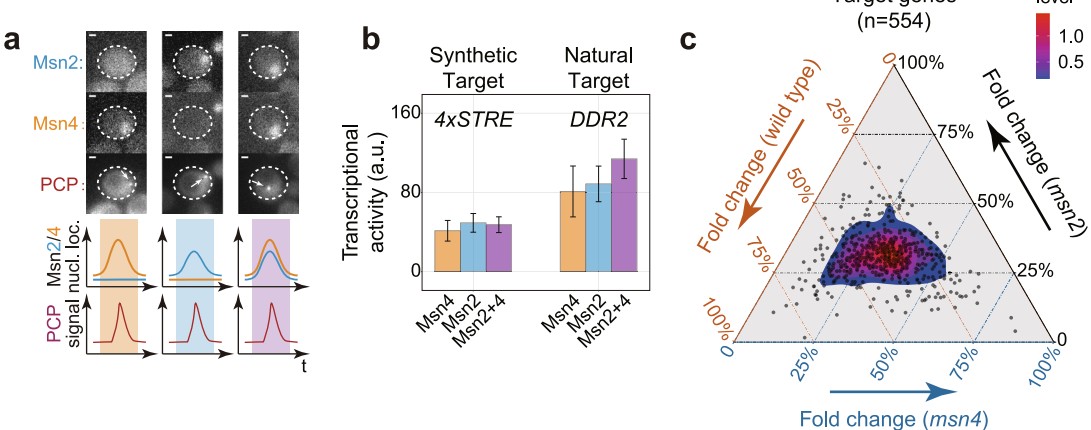

**Fig. 2 Individual pulses of Msn2 and Msn4 display functional redundancy. a** Example snapshots (top, out of 148, 227, and 262 events, respectively) and schematics (bottom) illustrating three types of TF nuclear localization dynamics (left to right: Msn4-only, Msn2-only, and coincident pulse) and target gene responses (i.e., nascent transcription sites indicated by white arrows). Scale bars indicate 1 μm. **b** Pulse-triggered averaging analysis[14] of the transcriptional activities of PP7-based synthetic reporter gene (*4xSTRE*) and endogenous target gene (*DDR2*) in response to three types of TF dynamics under glucose limitation stress (0.1% glucose). Event numbers are 148, 227, 262, 122, 247, and 260. Error bars indicate 95% CIs and centers indicate means. See also Supplementary Fig. 14a. **c** Genome-wide characterization of functional redundancy between Msn2 and Msn4 pulses by RNA-seq. Synchronized Msn4-only pulse, Msn2-only pulse, and Msn2/Msn4 co-pulse across cell population as in (**a**) were achieved by inducing cells of three different genotypes (*msn2*, *msn4*, and wild type) by 0.5 M KCl. For each target gene, expression fold change between stress versus non-stress conditions was calculated for each genotype and shown in the ternary plot after normalization. Colormap represents gene density.

redundant with respect to target gene activation. It is noted that the redundancy of these two TFs has not been explicitly analyzed at the single pulse level, especially during steady-state responses[31,32,35,38]. To do so, we implemented a single-cell assay to simultaneously monitor TF dynamics and real-time

transcriptional output of target genes, which was developed in our earlier work[14] (Figs. 1a and 2a).

More specifically, cells were stimulated with two different types of stresses (i.e., low glucose and high salt) to induce pulses of Msn2/Msn4, and multi-color time-series images were analyzed to

quantify target responses to Msn2/Msn4 pulses during steady-state responses. It is noted that target responses were analyzed at the level of nascent transcriptional activities with the PP7-based transcriptional reporter system, allowing us to capture target responses at a fast timescale. This strategy has been successfully implemented for understanding combinatorial gene regulation by pulsatile TFs[14].

Using pulse-triggered averaging analysis[14], we analyzed the responses of a synthetic Msn2/Msn4 target gene and an endogenous target *DDR2* to three possible scenarios of Msn2/Msn4 pulses, namely Msn2-only pulse, Msn4-only pulse, and Msn2/Msn4 co-pulse (Fig. 2a). The results showed that target responses to Msn2- and Msn4-only pulses are comparable to the responses to Msn2/Msn4 co-pulse, suggesting a large degree of functional redundancy at the level of individual Msn2/Msn4 pulses for both target genes under two stress conditions (Fig. 2b and Supplementary Fig. 4a). Note that under glucose limitation stress, *DDR2* exhibits a slightly stronger (but statistically insignificant) response under Msn2/4 co-pulse compared to that under Msn4- or Msn2-only pulses.

To analyze the redundancy at the single pulse level across the genome, we used RNA-sequencing (RNA-seq) to compare genome-wide target responses across three different strains, namely wild type, *msn2*, and *msn4*. More specifically, yeast cultures of wild type, *msn2*, and *msn4* were subjected to sudden salt stress (i.e., 0.5 M KCl) to trigger synchronized Msn2/Msn4 co-pulse (Supplementary Fig. 4b), Msn4- and Msn2-only pulse across cell population, respectively, and cells were collected at ~15 min post-stress addition (time around the end of the pulses) for RNA extraction and subsequent sequencing (Fig. 2c and Supplementary Fig. 4c). It is important to note that the population synchrony during transient responses allows us to quantify how cells respond to different types of pulses using bulk RNA-seq. The results showed that pulses of Msn2/Msn4 can redundantly activate a large fraction of their shared targets (~85%, which we term redundant target genes), as most of the target genes reside around the center of the ternary plots. Mechanistically, this could result from the scenario where each Msn2 or Msn4 pulse alone approaches or exceeds the saturation point of the dose–response curves for most of their target promoters, enabling an OR gate-like dynamic logic control of shared targets by Msn2/4 pulses. To test this hypothesis, we binned Msn2- or Msn4-only pulses based on the area of each pulse (defined in Supplementary Fig. 1b), and used the pulse-triggered averaging approach to compute the corresponding transcriptional activities of either the natural or the synthetic target gene (Fig. 2b). By doing so, we found that both target genes' transcriptional activities are relatively comparable across different bins of Msn2 or Msn4 pulses (Supplementary Fig. 4d), indicating that these two target genes' responses to both Msn2 and Msn4 are saturated at the single pulse level.

We further addressed whether redundant and nonredundant genes are enriched for different biological functions. By performing gene ontology analysis, we found that the top three enriched functions of redundant genes are "response to abiotic stimulus" ($p = 5.1e - 7$), "cellular response to chemical stimulus" ($p = 1.1e - 4$), and "response to osmotic stress" ($p = 2.7e - 4$). In contrast, the top three enriched functions of nonredundant genes are "regulation of transcription by RNA polymerase II" ($p = 1.1e - 5$), "negative regulation of transcription by RNA polymerase II" ($p = 1.5e - 4$), and "positive regulation of transcription by RNA polymerase II" ($p = 1.5e - 3$). Thus, it appears that these two types of genes are functionally different. Moreover, it would be intriguing to further investigate the activation kinetics of these genes and compare with the kinetics-based classification of Msn2/4 target genes[29–31].

**Temporal redundancy modulation leads to increased dynamic ranges of target expression**. Thus far, we have established that at the individual pulse level, Msn2/Msn4 have a high degree of functional redundancy with respect to target activation. In other words, pulses of Msn2 and Msn4 could activate most of the shared target genes in a redundant manner, yet for the rest of the target genes, pulses of Msn2 and Msn4 are not functionally redundant.

We next speculated how different types of target genes could be regulated by the modulation of Msn2 and Msn4 pulses in response to changes in stress level. We postulated that, by decreasing Msn2/Msn4 pulse coincidence rate, cells effectively "separate" the coincident (and redundant) Msn2/Msn4 pulses to enhance gene activation capacity for redundant targets (Fig. 3a). In contrast, nonredundant targets would be affected by changes in pulse frequency but not by the change in pulse coincidence rate. And, because in all stresses we tested (Fig. 1c and Supplementary Fig. 3a), we found that stress level affects the pulse coincidence rate in addition to the pulse frequencies. We thus hypothesized that cells could achieve a larger dynamic range of activation for the redundant targets of Msn2/Msn4 when compared to the nonredundant targets in response to changes in stress level.

To test this hypothesis, we carried out steady-state transcriptomic analysis of yeast cells under a gradient of glucose limitation stresses (the same stress conditions as Fig. 1c). Under these conditions, the temporal relationship between Msn2 and Msn4 dynamics at steady state is modulated by stress level, i.e., their pulse coincidence rate decreases as stress level increases (Fig. 1c). By taking a "snapshot" of the steady-state populational transcriptomic profile using RNA-seq, we would be able to capture the averaged transcriptional responses of Msn2/4 target genes in cells whose Msn2/4 dynamics are regulated as in Fig. 1c. The rationale is that messenger RNA (mRNA) half-lives are short compared to the steady-state measurement (~4 h) and an analogous approach was used previously to quantify target responses to pulsatile TFs in yeast[14]. By doing so, we could test the aforementioned hypothesis regarding how the temporal relationship between Msn2 and Msn4 might be controlled to regulate target genes. As expected, the dynamic range of the redundant targets significantly exceeds that of the nonredundant targets, supporting the idea that the temporal relationship between Msn2 and Msn4 pulses is modulated to enhance the dynamic range of redundant target genes under a gradient of stresses (Fig. 3b and Supplementary Fig. 4e). This result is consistent with the picture that the activation of redundant target genes is saturated at the level of individual Msn2/4 pulses (Supplementary Fig. 4d). Furthermore, it is noted that because expression fold change was used to quantify dynamic range, the observed difference in dynamic range would not result from the difference in transcript half-lives.

Therefore, it appears that stress could regulate the redundant targets by modulating the degree of redundancy between these two paralogs in a temporal manner. In this model, by altering the temporal relationship between functionally redundant Msn2 and Msn4 pulses, cells could dynamically modulate the degree of redundancy between Msn2 and Msn4, resulting in an effective modulation of redundancy in a temporal manner. We thus termed this mode of gene regulation as temporal redundancy modulation.

**Temporal redundancy modulation provides fitness advantages for cells under stress**. While temporal redundancy could be modulated to upregulate stress response genes (that are redundantly regulated), it is unclear whether cells could gain fitness

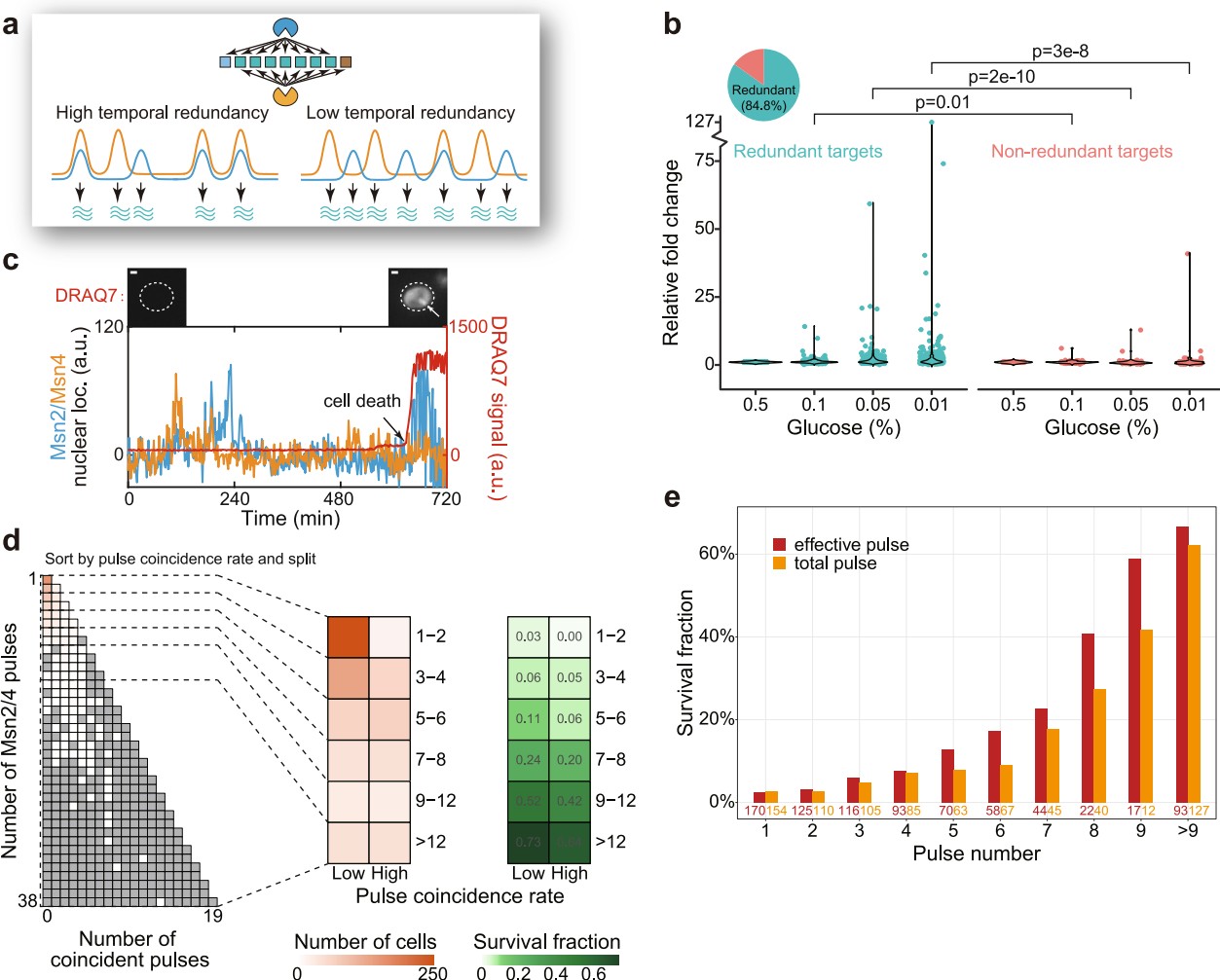

**Fig. 3 Temporal redundancy modulation leads to increased dynamic ranges of target expression and can enhance stress survival capacity. a** Schematics showing that cells could modulate temporal redundancy to control the expression of redundant target genes (cyan squares on the top). (Bottom) While pulse frequencies remain unchanged, the two TFs display a lower degree of temporal redundancy on the right, leading to more mRNA expression for the redundant targets (cyan squiggly lines). **b** Temporal redundancy modulation contributes to increased dynamic ranges of target genes. Steady-state gene expression levels under four different glucose limitation stresses are shown for both redundant and nonredundant targets of Msn2/Msn4. Redundant targets show elevated activation fold changes compared to nonredundant targets at high stress conditions, presumably due to the decrease in temporal redundancy at high stress levels (Fig. 1c). Gene number is 436 and *p* values indicate one-sided Kolmogorov–Smirnov (K–S) test. **c** Single-cell assay to quantify the physiological effect of temporal redundancy modulation on yeast stress survival. In this example cell (out of 1676 cells), Msn2 (blue) and Msn4 (orange) nuclear localization dynamics, together with the signal from a live-dead reporter DRAQ7 (red), were recorded under glucose limitation stress (0.05% glucose). Scale bars indicate 1 μm. See also Supplementary Movie 2. **d** Quantifying the effect of temporal redundancy modulation on stress survival. Cells were grouped based on similar total Msn2/4 pulse numbers during the first 7-h movie under glucose limitation stress (0.05% glucose) and within each group, cells were divided into two subgroups based on the pulse coincidence rate in each cell (see "Methods" for details). Note that cells were binned to increase the cell number for the calculation of survival fraction. Survival fraction after 6 more hours for each subgroup was then calculated. Eight hundred and eight cells that survived >7 h (out of 1676 cells) were analyzed in this figure. **e** Survival fractions for cells with varying numbers of effective or total Msn2/4 pulses. Cells in (**d**) were grouped based on the number of effective pulses or total Msn2/4 pulses, and the survival fraction of each group was calculated. Cell numbers are indicated in the graph.

advantages. To test this, we developed an assay to measure the temporal relationship between Msn2/4 dynamics and the stress survival ability simultaneously at the single-cell level, allowing us to decipher the potential role of temporal redundancy modulation during stress response.

In this assay, cells with fluorescent protein-tagged Msn2/Msn4 are subjected to the low glucose stress, and the DRAQ7 dye is added to the medium so that the nuclei of dying cells are stained and emit infrared fluorescence (Fig. 3c and Supplementary Movie 2). In doing so, we continuously monitored Msn2/Msn4 dynamics and cell fate changes in the same single cells, allowing

us to quantify the relationship between pulse coincidence rate and stress survival capability on a cell-by-cell basis. It is noted hereby that cell fate refers to whether cells would survive the stress. We then carried out multiple analyses to investigate the potential role of temporal redundancy modulation in stress survival.

In the first analysis, we aimed to compare the survival capability between cells with comparable total pulse numbers of Msn2 and Msn4, but with different pulse coincidence rates. We thus binned cells having similar total pulse numbers, and obtained two subgroups of cells for each binned group based on the pulse coincidence rate in each cell (see "Methods" for

details). By doing so, we found that for the six different groups of cells (containing varying total pulse numbers), the subgroups with lower pulse coincidence rates all have higher stress survival rates compared to their counterparts (Fig. 3d). This result is consistent with our hypothesis that cells with more noncoincident pulses can achieve higher expression levels of redundant target genes that are important for stress survival (see Fig. 3a for a schematic).

In the second analysis, we tested whether the cell survival capacity negatively correlates with the number of coincident pulses in a statistically significant manner. We thus constructed a linear regression model with two independent variables, including the total pulse number and the number of coincident pulses between Msn2 and Msn4 pulses. We found that indeed the cell survival rate negatively correlates with the number of coincident pulses (Supplementary Fig. 5b). We further compared this model with a second linear regression model that contains only the total pulse number as the variable. We found that the first model can explain the variation in cell survival better than the second model (Supplementary Fig. 5b), consistent with a picture that the temporal relationship between Msn2 and Msn4 pulses is a key parameter regulating stress survival.

We next carried out additional analyses by comparing the stress survival roles of effective pulse numbers with total pulse numbers. Because Msn2 and Msn4 are largely redundant at the single pulse level, each Msn2/Msn4 coincident pulse is as effective as an Msn2/Msn4-only pulse. Thus, based on the above results, we hypothesized that cells with a specific number of effective pulses would have a higher survival capability compared to cells with the same number of total pulses, because there are coincident pulses in the latter group, which would reduce the effective pulse number. To test this, we calculated the survival rates for groups of cells with different effective pulse numbers and with corresponding total pulse numbers. We found that with the same number of effective pulse or total pulse, the former group of cells has a higher stress survival capacity for almost all the pulse numbers analyzed (Fig. 3e). In addition, stress survival rate increases monotonically with either effective or total pulse number as expected. Furthermore, we predicted that adding an extra effective pulse would likely yield a larger increase in cell survival capability compared to adding an extra total pulse. This is because that an extra effective pulse (by definition) would lead to additional expression of redundant target genes, while an extra total pulse might not. To test this, we computed the fold change between the cell survival contributions by a single effective pulse and by a single total pulse. We found that under most scenarios, adding an extra effective pulse indeed results in a larger contribution to cell survival compared to adding an extra total pulse (Supplementary Fig. 5c).

Critically, the consistently higher stress survival capacity in cells with lower pulse coincidence rates, the negative correlation between coincident pulse number and survival capacity, as well as the contrast in survival capacity between cells with the same number of effective versus total pulses, all depended on the transcriptional activity of Msn2/Msn4 pulses, as these relationships were abolished in cells containing Msn2/Msn4 mutants (i.e., without DNA-binding domains; Supplementary Fig. 5d, e). It is noted that in this mutant strain, the stress survival rate appears to be higher compared to the strain with wild-type Msn2/Msn4 under our experimental condition. While this phenomenon cannot be accounted for by our model and warrants further investigations, it does not contradict with our model.

To ensure that the enhanced cell survival ability resulted from the upregulation of the redundant targets of Msn2/Msn4 (Fig. 3a), we performed analogous experiments with two mutant strains lacking either Msn2 or Msn4. We observed that increasing the

frequency of either Msn2 or Msn4 pulse led to enhanced stress survival (Supplementary Fig. 6a, b), suggesting that the redundant targets likely accounted for the apparent gain in stress survival ability for wild-type cells with lower pulse coincidence rates (Fig. 3d). Note that increasing the area, duration, or amplitude of either Msn2 or Msn4 pulse did not lead to enhanced cell survival, and increasing the accumulation of nuclear Msn2 or Msn4 did not lead to enhanced cell survival as significantly as increasing pulse frequency alone (Supplementary Fig. 6c), which is consistent with the picture that redundant target genes are largely saturated by individual Msn2 or Msn4 pulses.

Intriguingly, we found that the pulse coincidence rate between Msn2 and Msn4 decreases along the long time course under glucose limitation stress (Supplementary Fig. 6d), indicating that cells appeared to actively modulate the pulse coincidence rate. This observation is consistent with a scenario that the stress level increased with time, which might result from both the decrease in glucose concentration and the photo-toxicity, and thus cells would reduce the pulse coincidence rate as our model suggests. To further investigate this observation, we implemented a different assay with much larger cell numbers under either low (0.05%) or high (2%) glucose concentrations. In these experiments, instead of tracing individual cells by taking frequent snapshots, we took one frame per hour and quantified the pulse coincidence rate of surviving cells at each time point. Results showed that cells lowered their pulse coincidence rates at later time points only when they are under low glucose condition (Supplementary Fig. 6e), consistent with the picture that the stress level might increase over time.

Taken together, our results demonstrated that budding yeast cells could modulate the temporal relationship between redundant Msn2/Msn4 pulses to upregulate the expression of redundant stress response genes, which contributes to increased fitness levels under stress. In other words, temporal redundancy modulation of Msn2 and Msn4 could enhance yeast stress survival.

**Molecular circuitry for temporal redundancy modulation**. Protein kinase A (PKA) is a key regulator of Msn2/Msn4 activity[9,39]. To understand the pulsing of Msn2/Msn4, a model was recently proposed where TF pulses are driven by stochastic downward pulses of PKA activity[40]. To account for the divergent control of Msn2/Msn4 dynamics, we hypothesized that the paralogs might have evolved differential affinities to PKA. To test this hypothesis, we constructed two mutant strains carrying Msn2 or Msn4 only, which are driven by the same promoter (Supplementary Fig. 7a), and carried out multiple assays to compare their regulations by PKA.

In the first assay, we tested the prediction that if Msn2 and Msn4 possess differential affinities to PKA, then their nuclear localization responses to PKA activity inhibition would be different. More specifically, the TF with a lower affinity to PKA would be dephosphorylated faster, resulting in a more rapid shuttling into the nucleus if the TF initially is phosphorylated by PKA and thus localizes in the cytoplasm. Using the two mutant strains, we measured the nuclear localization responses of cytoplasmic localized Msn2-CFP and Msn4-CFP upon the addition of H-89, a validated PKA inhibitor for budding yeast[41]. The result showed that the PKA inhibitor H-89 induced a much more rapid cytoplasm-to-nucleus shuttling of Msn2-CFP compared to Msn4-CFP (Supplementary Fig. 7b), suggesting that under the same inhibited PKA activity, Msn2 has a lower affinity to PKA and is much more easily dephosphorylated compared to Msn4. Importantly, the observed difference in the affinities to PKA cannot be explained by the difference in the protein levels

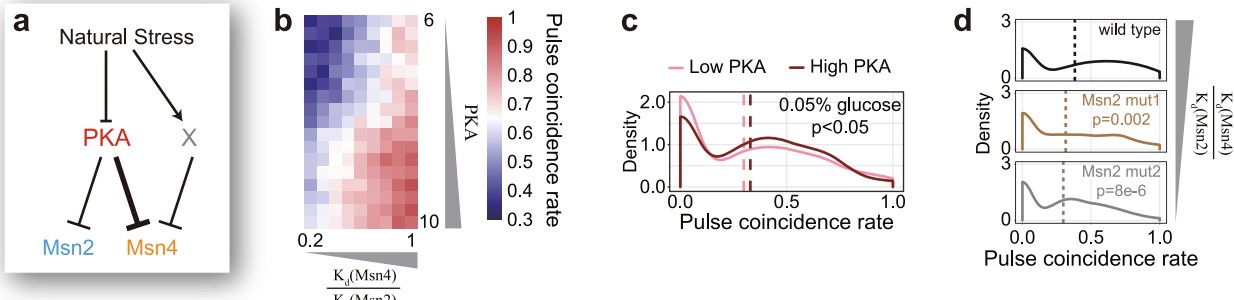

**Fig. 4 Dissecting the mechanism for temporal redundancy modulation. a** A proposed circuitry mechanism for achieving divergent control of Msn2 and Msn4 dynamics (see also Supplementary Fig. 9a). **b** Stochastic simulations of the model revealed intrinsic control (e.g., differential affinities to PKA) and extrinsic control (e.g., PKA activity, which is stress-dependent) of the temporal relationship between Msn2 and Msn4 dynamics. **c** Experimental validation for the extrinsic control of temporal redundancy through artificially altering PKA level (Supplementary Fig. 10a). Elevating PKA increases pulse coincidence rate, as the model predicted (Supplementary Fig. 9d, f). Dashed lines indicate means and the $p$ value indicates one-sided K–S test. Cell numbers are 513 and 333 for low (0.25 μM $Cu^{2+}$) and high (1 μM $Cu^{2+}$) PKA. **d** Experimental validation for the intrinsic control of temporal redundancy. Two mutants of Msn2 were generated to tune the relative affinity to PKA between Msn2 and Msn4 (Supplementary Fig. 10b, c) and their pulse coincidence rates were compared with wild type under glucose limitation stress (0.01% glucose, see also Supplementary Fig. 10d). Dashed lines indicate means and $p$ values indicate one-sided K–S tests. Cell numbers are 363, 487, and 503 (top to bottom).

(Supplementary Fig. 7c, d). Based on this result, we predicted that cells would induce a weaker Msn4 pulse compared to Msn2 during the transient responses to diverse stresses. This is because that Msn4 would be more easily phosphorylated by PKA, and thus more easily exported out of the nucleus compared to Msn2. Using the above two mutant strains, the results indeed showed that under all three stresses tested, the transient pulse of Msn4-CFP had either a lower amplitude or a shorter duration compared to that of Msn2-CFP (Supplementary Fig. 7e–g). Together, these results provide multiple lines of evidence supporting that Msn2 and Msn4 have differential affinities to PKA, which likely contributes to their divergent pulsatile dynamics.

Furthermore, since Msn4, but not Msn2, exhibits pulsatile dynamics in the absence of PKA[28], we hypothesized the presence of a second kinase (kinase X) regulating the activity of Msn4, thus constituting an incoherent feedforward regulation[42] (Fig. 4a) that could explain the biphasic dependence of Msn4 frequency on stress level (Supplementary Fig. 8a). To identify kinase X, we first searched for potential post-translational regulators of Msn2 or Msn4 (see Supplementary Fig. 8b for the list). We then constructed copper-inducible strains for tuning the expression levels of these potential regulators and conducted time-lapse measurements of these strains under different copper levels to quantify the impact of increasing regulator concentration on Msn2/4 pulse frequencies. We found that when elevating Yak1 level, Msn4 frequency decreased, while Msn2 frequency did not change (Supplementary Fig. 8c, d). We further carried out a different assay with much larger cell numbers to show that the percentage of cells with nuclear localization of Msn4 decreased, while that of Msn2 did not change when increasing Yak1 level (Supplementary Fig. 8e). These results indicated that Yak1 may be a candidate for kinase X, supporting our proposed circuit mechanism (Supplementary Fig. 9a).

To provide mechanistic insights into temporal redundancy modulation, we constructed a mathematical model that incorporates the differential phospho-regulation on Msn2 and Msn4, and performed stochastic simulations of the model (Supplementary Fig. 9b, c). Results from simulations qualitatively recapitulated key features of the experimental single-cell data, including the divergent pulsatile dynamics of Msn2 and Msn4 (Supplementary Fig. 9c), as well as the bimodality in the pulse coincidence rate (Supplementary Fig. 9d, e). Using this model, we revealed both extrinsic and intrinsic modulations of temporal redundancy

(Fig. 4b). First, model simulations predicted that an increase in PKA level would lead to an increase in the pulse coincidence rate (i.e., extrinsic modulation, Fig. 4b and Supplementary Fig. 9d, see Supplementary Fig. 9f for additional analysis of the parameters). By artificially expressing different levels of PKA catalytic subunits (Supplementary Fig. 10a), we showed that elevating PKA resulted in a change in pulse coincidence rate, as well as a corresponding shift in its distribution (Fig. 4c), qualitatively matching the model predictions. Second, the model also predicted that an increased difference in the affinities to PKA between Msn2 and Msn4 would lead to a decrease in their pulse coincidence rate (i.e., intrinsic modulation, Fig. 4b and Supplementary Fig. 9e, see Supplementary Fig. 9g for additional analysis of the parameters). To test this, we constructed two Msn2 mutants that displayed lower affinities to PKA than wild-type Msn2 (Supplementary Fig. 10b, c), which effectively increases the difference in the affinities to PKA between Msn2 and Msn4. We found that the pulse coincidence rates between these two Msn2 mutants and wild-type Msn4 are modulated in a manner that agrees with the model prediction (Fig. 4d and Supplementary Fig. 10d). Note that the decrease of pulse coincidence rate in mutants was not due to inactivation of Msn2 (Supplementary Fig. 10c).

Thus, we conclude that the differential affinities to PKA for Msn2 and Msn4, together with the additional regulation of Msn4 by a separate kinase, could enable the modulation of temporal redundancy between Msn2 and Msn4. However, due to the limited screen we performed, we might have missed additional regulatory mechanisms that are important for the modulation of Msn2/4 pulses. That said, while our model may not be the only model for explaining the divergent Msn2/4 dynamics, it captures the main experimental results and thus sheds a light on how the divergent temporal dynamics of paralogous TFs could be achieved via post-translational mechanisms (see Supplementary Fig. 9a for discussions about alternative models).

**A potential evolutionary role of temporal redundancy modulation.** Unexpectedly, after ~100 million years of post-WGD evolution, functional redundancy between many paralog pairs, including Msn2 and Msn4, has been preserved in budding yeast[43–45]. Similar observations were made in *Caenorhabditis elegans* and other organisms[46,47], and models have been proposed to account for such evolutionary stability of paralog redundancy.

In these models, redundancy was thought to be maintained by mechanisms such as co-selection on nonoverlapping divergent functions[46], dosage compensation[48], or *cis*-regulatory divergences[35,49–53] that confer selective advantages. These mechanisms imply a general constraint—the preservation of a certain degree of functional redundancy requires the paralogs to diverge at the level of either protein function or *cis*-regulation (to yield selective advantages).

We next discussed the potential role of temporal redundancy modulation for the evolution of paralogs. Because temporal redundancy modulation could confer fitness advantages for yeast cells under stress, i.e., cells could exploit temporal interactions between functionally redundant paralogs to increase their fitness levels, we postulated that such a mechanism might, in turn, help the preservation of redundant paralogs. In other words, functionally redundant paralogs could be fixed in evolution if (1) they both exhibit activity dynamics, and (2) their dynamic relationship can be exploited together with their functional redundancy to increase organismal fitness. Importantly, compared to the aforementioned models, our hypothesis focuses on the temporal activity dynamics of the paralogs, instead of their biochemical function or *cis*-regulation.

To test the hypothesis, we first determined whether the ancestral Msn2/Msn4 acquired pulsatile dynamics prior to WGD. To do so, we examined the orthologs of Msn2/4 in two related yeast species. *Kluyveromyces lactis* is a frequently used pre-WGD reference species[34,54]. By endogenous fluorescent protein tagging and time-lapse microscopy, we found that Msn2/4 ortholog in *K. lactis* exhibits nuclear-cytoplasmic shuttling dynamics under low glucose stresses (Supplementary Fig. 11a–c and Supplementary Movie 3). In contrast, the Msn2/4 ortholog in *Schizosaccharomyces pombe* (a species that is much more distant to budding yeast and did not undergo WGD) is a specific regulator of oxidative stress response[55] and displays no apparent activity dynamics under oxidative stresses (Supplementary Fig. 11d, e and Supplementary Movie 4).

Together, these results suggest that the ancestral Msn2/Msn4 acquired pulsing before WGD. Thus, after duplication, the two paralogs needed to evolve certain mechanisms in order to maintain their functional redundancy over a long timescale. In our hypothesis, a mechanism for modulating their temporal relationship would help to preserve their functional redundancy.

**Evidence for the potential evolutionary role of temporal redundancy modulation**. To further test the hypothesis, we next analyzed a related yeast species *C. glabrata* that diverged from *S. cerevisiae* soon after WGD[56] (Fig. 5a). Because Msn2/Msn4 orthologs in *C. glabrata* had also evolved for a long time, we wondered whether this species also preserved their pulsatile dynamics and functional redundancy in target activation, and at the meantime, evolved a mechanism to modulate the temporal relationship between their dynamics.

Through fluorescent protein tagging and single-cell imaging, we found that the Msn2/Msn4 orthologs in *C. glabrata* display a highly tunable temporal relationship (Supplementary Fig. 12a, b), and their pulse coincidence rate is modulated by stress levels in a manner that is similar to budding yeast (Fig. 5b). At the single-cell level, the pulse coincidence rate also displays bimodality (Fig. 5c) and the bimodal distribution is also stress level-dependent (Supplementary Fig. 12c). To quantify the degree of functional redundancy genome-wide between the paralogs, we first identified shared target genes of Msn2_cg/Msn4_cg through differential gene expression analysis (Supplementary Fig. 12d). We then used a similar assay as above to synchronously induce three types of pulses, namely Msn2_cg-only, Msn4_cg-only, and

Msn2_cg Msn4_cg co-pulse, and compared the induction levels of the shared targets. Intriguingly, most target genes are redundantly activated by the two paralogs at the single pulse level, indicating a significant functional redundancy between Msn2_cg and Msn4_cg (Fig. 5d and Supplementary Fig. 12e), as in budding yeast.

Thus, both *S. cerevisiae* and *C. glabrata* preserved the temporal dynamics of Msn2 and Msn4 as well as their functional redundancy, and appear to have evolved mechanisms for tuning their temporal relationship, which is consistent with the picture that temporal redundancy modulation might have contributed to the preservation of functional redundancy during the post-WGD evolution of these two species (Fig. 5e). Intriguingly, in *C. glabrata* the two paralogs display a larger degree of divergence between their temporal dynamics while maintaining more functional redundancy (based on the fraction of redundant target genes) compared to *S. cerevisiae*. Such relationships further support our hypothesis that the divergence in temporal dynamics could help to maintain functional redundancy between paralogs. It is important to note that the mechanism for temporal redundancy modulation could have been evolved before the two species diverged, which would also be consistent with our hypothesis.

We next investigated the potential evolutionary role of temporal redundancy modulation more broadly. We reasoned that if temporal redundancy modulation is a common mechanism for retaining functional redundancy for TF paralogs, then the TFs with redundant paralogs would be more likely to display temporal activity dynamics compared to the ones without paralogs, because temporal activity dynamics of the paralogs are the prerequisite for temporal redundancy modulation. To test this, we needed to analyze single-cell datasets that characterize the temporal activity dynamics of many TFs.

Using imaging-based proteome-wide screen, it was previously found that TFs in *S. cerevisiae* appear to be enriched for paralogs that display pulsatile dynamics[25]. Statistical analysis confirmed that TFs with redundant paralogs are more likely to exhibit pulsatile dynamics than the ones without paralogs (Supplementary Fig. 13a), consistent with our model prediction. We further examined this association in human cells. Since paralogous TFs in human cells tend to exhibit functional redundancy (Supplementary Fig. 13b), we asked whether these TFs also display more activity dynamics compared to the singleton TFs. Using public single-cell ATAC-seq (Assay for Transposase-Accessible Chromatin using sequencing) data[57] (Supplementary Fig. 13c), we found that paralogous human TFs are indeed more dynamic than the singletons across six different cell lines (Supplementary Fig. 13d, e).

Thus, data from both yeast and human cells showed that TFs with redundant paralogs are more likely to exhibit temporal activity dynamics, consistent with our hypothesis that temporal redundancy modulation appears to play a role in the evolution of TF paralogs (Fig. 5f).

### Discussion

In summary, by combining both experiments and mathematical modeling, we have shown that temporal redundancy modulation of the general stress response regulators, Msn2 and Msn4, represents a dynamics-based mechanism for cell fate control in budding yeast. To offer a systematic view of temporal redundancy modulation, we quantified how cells encode environmental inputs into the temporal relationship between the paralogs, how downstream targets decode the dynamics of the paralogs, and how such dynamics-based signal encoding and decoding affects cell physiology during stress response. Furthermore, we provided several lines of evidence supporting the potential role of temporal

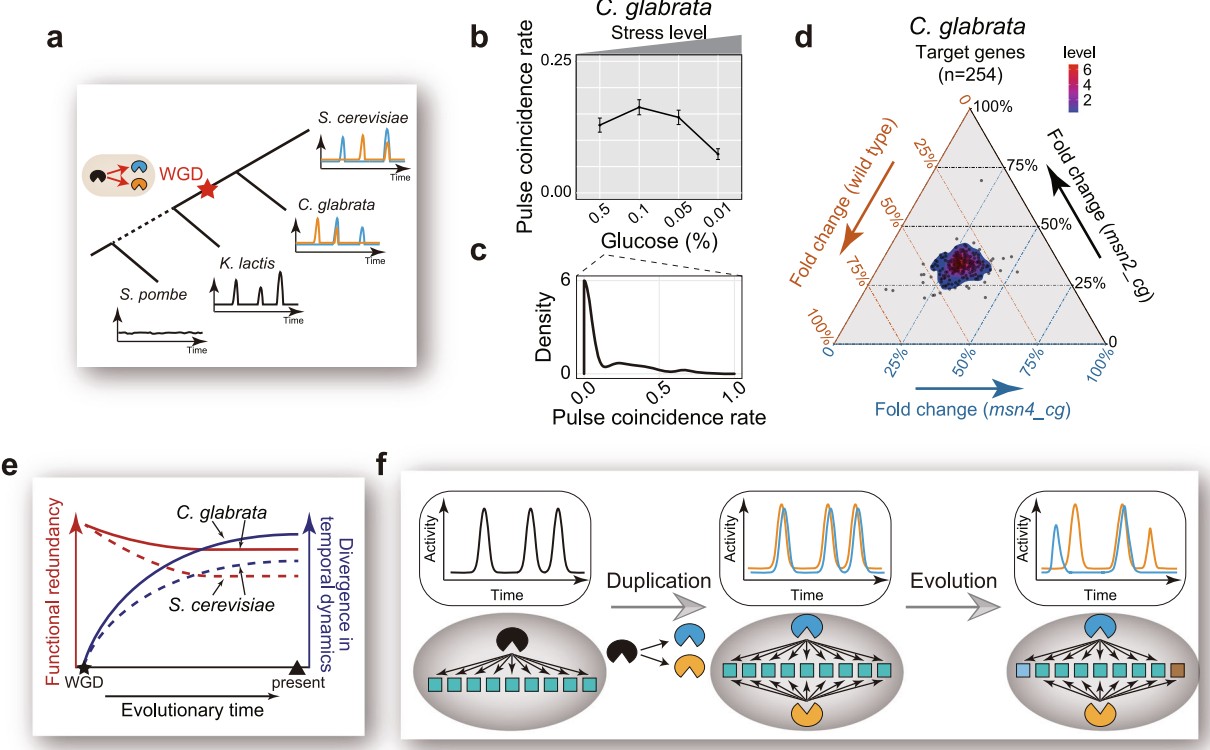

**Fig. 5 Evidence supporting the evolutionary role of temporal redundancy modulation. a** Evolution of pulsatile dynamics of Msn2/4 orthologs in different yeast species. See also Supplementary Fig. 11. **b** Pulse coincidence rate between Msn2_cg and Msn4_cg in *C. glabrata* decreases as the glucose limitation stress increases. Cell numbers are 525, 652, 761, and 752. Error bars indicate 95% CIs and centers indicate means. **c** Pulse coincidence rate is bimodally distributed among single cells under the same stress level for *C. glabrata*. **d** Quantification of functional redundancy between Msn2_cg and Msn4_cg in *C. glabrata* through analogous experiments and analysis as in Fig. 2c. Msn2/4_cg target genes were identified by differential gene expression analysis (Supplementary Fig. 12d). **e** Proposed evolutionary trajectories of functional redundancy and temporal relationship between paralog dynamics for the Msn2/4 system in yeast. **f** Schematics illustrating the potential role of temporal redundancy modulation for the evolutionary preservation of paralogous redundancy. Prior to duplication (leftmost panel), the TF of interest exhibits pulsatile dynamics (top) and regulates a set of target genes (bottom). Immediately after duplication, paralogous TFs have high functional redundancy and highly correlated dynamics (middle panel). The subsequent acquisition of mechanisms for temporal redundancy modulation helps the maintenance of functional redundancy (rightmost panel). For each panel, the top half illustrates the temporal activity dynamics of the TF(s), while the bottom half illustrates the regulatory connections between TF(s) and targets. Colored squares represent shared or unshared targets of the TFs.

redundancy modulation in the evolutionary preservation of functionally redundant TF paralogs. We note that while our temporal dynamics-based hypothesis for explaining the preservation of redundant paralogs is distinct from existing hypotheses based on divergence in *cis*-regulation or protein function, it might likely complement rather than contradict with these models. Yet, the universality of our proposed mechanism requires further investigations using single-cell approaches.

More generally, after the duplication of a dynamic TF into two paralogs, cells could evolve a mechanism to modulate their temporal relationship by acquiring differential post-translational regulation, as we demonstrated for the Msn2/Msn4 system in budding yeast. Through extrinsic control mechanisms, cells could exploit the temporal relationship between paralogs, in conjunction with their functional redundancy, to increase fitness levels by upregulating redundant target genes, which could, in turn, help the maintenance of functionally redundant paralogs. While the exact circuit-level mechanism for achieving temporal redundancy modulation in our system requires further investigations, our proposed post-transcriptional circuit sheds a light on how the temporal dynamics of two paralogs could be divergently controlled to yield potential physiological functions. It should be noted that the modulation of paralog dynamics could also be achieved by transcriptional circuits, and might confer other types

of physiological advantages for the cell, such as enhancing the signal processing capacity in the mammalian NFAT1/NFAT4 system[21].

Besides offering some insights into the evolution of paralogs and biological robustness, our results also suggest an evolutionary force driving the potentially widespread prevalence of signaling pathways with dynamic paralogs, underscoring the essential roles of temporal dynamics in cellular regulation[4,5]. Given the importance of signaling dynamics in both healthy and diseased cells[22,58], it is tempting to speculate that temporal dynamics such as pulsing may be exploited in the evolution of diseases such as cancer. More generally, since temporal dynamics are often induced by biological noise or fluctuations[40], our work also highlights the critical role of noise in the evolution of genetic circuits, and the continuing growth of large-scale single-cell datasets will allow further dissecting the functional roles of noise-induced temporal dynamics in microbes and other organisms[5].

## Methods

**Strain construction**. Molecular cloning was performed according to the manufacturer's protocols. Plasmids were replicated in DH5α *Escherichia coli*. All *S. cerevisiae* strains were constructed based on BY4741, all *Schizosaccharomyces pombe* strains were constructed based on 972h- (a gift from Chao Tang lab), all *K. lactis* strains were constructed based on CBS 141 (acquired from China General Microbiological Culture Collection Center, collection number: CGMCC 2.1494), all

*C. glabrata* strains were constructed based on BG2 (a gift from Ping Wei). All transformations were performed with Frozen-EZ Yeast Transformation II Kit (Zymo Research). Results of transformations were confirmed by PCR and/or sequencing. All strains are listed in Supplementary Table 1. All unique materials used are readily available from the authors.

For endogenous gene deletion or fusion with fluorescent proteins, PCR fragments containing upstream homology arm (~400 bp, before start codon for gene deletion or before stop codon for gene fusion), fluorescent protein gene (only for gene fusion), selective marker cassette, and downstream homology arm (~400 bp, after stop codon) were constructed by fusion PCR and then transformed into target strains. For *TEF1* promoter-driven RNA-binding protein fusion PCP-iRFP, the fragment was inserted to the *LYS2* locus. For endogenous gene fusion with *24xPP7* cassette, fragments containing upstream homology arm (~400 bp, before stop codon with stop codon preserved), *24xPP7* cassette, selective marker, and downstream homology arm (~400 bp, after stop codon) were obtained through plasmid cloning followed by enzyme digestion. For synthetic promoter-driven (*4xSTRE*, sequence: 4× gatctacagcccctggaaaat) *24xPP7*, the fragments were inserted to the *TRP1* locus. The sources for all the cassettes used are: *CFP* (mTurquoise2[59]), *YFP* (yECitrine from pKT0139 or yEVenus from pKT0090), *iRFP* (iRFP713[60]), *LEU2* (from pRS315 plasmid), *spHIS5* (from pKT355 plasmid), *CaURA3* (from pKT174 plasmid), *MET17* (from pHLUM plasmid), *KanMX* (from pKT0103 plasmid), *hphMX6* (from pAG32 plasmid), *natMX6* (from pAG25 plasmid), *PCP* (from pDZ276 plasmid), *24xPP7* (from pCR4-24xPP7SL plasmid). For copper-inducible strains, the endogenous promoters were replaced by *CUP1-1* promoter (−291 to −1 of the *CUP1-1* gene).

For mutations of Msn2 and Msn4, fragments with upstream and downstream homology arms, Msn2 (or Msn4) coding sequence with desired mutations, fluorescent protein, and selective marker were transformed to replace the endogenous targets. For the deletion of DNA-binding domain of Msn2, amino acids (a.a.) 642–704 were deleted. For the deletion of DNA-binding domain of Msn4, a.a. 568–630 were deleted. For mutations of PKA consensus sites in Msn2, (R/K)(R/K)XS was defined as PKA consensus motif and the PKA consensus site a. a. 617–620 and a.a. 622–625 were mutated to RRXA (Msn2 mut1: *Δaa617–620:: RRPA*, *Δaa622–625::RRKA*) or deleted (Msn2 mut2: *Δaa619–629::PVQPRK*).

To explore the evolutionary role of temporal redundancy modulation, orthologs of Msn2 and Msn4 in related yeast species (*S. pombe*, *K. lactis*, and *C. glabrata*) were identified through phylogenetic analysis, namely, HSR1 in *S. pombe*[61], KLLA0F26961g in *K. lactis*[62], CAGL0F05995g (orthologous to Msn2 or Msn2_cg) and CAGL0M13189g (orthologous to Msn4, or Msn4_cg) in *C. glabrata*[62]. Molecular cloning for fluorescent protein labeling or gene deletion in these strains was carried out similarly as in *S. cerevisiae*. Primers used in this study are listed in Supplementary Table 2.

**Media and growth conditions**. For the transformation of all strains except *S. pombe*, 2xYPD medium was used, which is composed of 2% (w/v) yeast extract (OXOID), 4% (w/v) Bacto peptone (BD), and 4% (w/v) glucose (Sigma). For the transformation of *S. pombe*, YE medium was used, which is composed of 0.5% (w/v) yeast extract and 3% (w/v) glucose. For microscopy and other experiments of all strains except *S. pombe*, a synthetic media with low autofluorescence was used. More specifically, yeast nitrogen base w/o amino acids (BD), amino-acid drop-out mix (Clontech or MP), and Milli-Q water were mixed accordingly, and glucose was then added to 2% (w/v) for normal culture. For microscopy of *S. pombe*, EMM2 media (Elite Media) were used. Detailed information of reagents used in this study is listed in Supplementary Table 3.

For microscopy, single colonies were picked from agar plates the day before and the overnight culture was diluted to OD$_{600}$ = 0.1 with a total of 2 mL in glass tubes 4 h prior to imaging. For quantitative polymerase chain reaction (qPCR) and RNA-seq experiments, overnight culture was diluted to OD$_{600}$ = 0.25 (for steady-state experiments) or OD$_{600}$ = 0.5 (for transient stress experiments) with a total of 10 mL in 50 mL centrifuge tubes (NEST). All strains were cultured at 30 °C under shaking (220 rpm).

**Time-lapse microscopy**. All time-lapse microscopy was performed on a Nikon Ti-E microscope with a 100× Plan Apo Lambda objective and Perfect Focus System (PFS, Nikon). Fluorescence was excited by an LED light source (Solar, Lumencor) and collected to an sCMOS camera (Hamamatsu, C13440-20CU) with 2-by-2 bin. For imaging PCP-iRFP, a three-slice z-stack with 1 μm spacing was acquired. For imaging DRAQ7 dye (Abcam) in survival experiments, the filter was the same as that of iRFP. The excitation and emission filters for CFP, YFP, and iRFP are: Ex 436/20 and Em 480/40, Ex 500/20 and Em 535/30, and Ex 650/45 and Em 720/60, respectively. The frame rate was one frame per 2 min, unless otherwise specified. Automated microscopy was conducted with Micro-Manager[63] and the temperature was maintained at 30 °C with a customized heating chamber.

During imaging, cells were cultured either in 96-well plates (Cellvis), agarose pads, or microfluidic devices. When conducting microscopy with 96-well plates, 2 mg/mL concanavalin A (Sigma) solution was incubated for ~8 min prior to cell incubation. Microscopy with agarose pads was conducted according to the protocol of Young et al.[64]. The microfluidic device was modified from the design of Zhang et al.[36] (Supplementary Fig. 1a). The microfluidic device was fabricated with polydimethylsiloxane and bonded with a 24 mm × 50 mm glass coverslip after air-

plasma cleaning (Harrick Plasma). Cells were briefly centrifuged to concentrate before loading into microfluidic channels. Media flow was controlled by syringe pumps (LONGER, LSP04-1A) with 10 mL syringes (BD) and the flow rate was 100 μL/h. The starting condition was 2% (w/v) glucose and then microscopy began for ~40 min before cells were subjected to the stress conditions. The switch from normal to stress-containing media was performed manually and the stress arrival time was identified through manual visualization of synchronous nuclear localization of Msn2 (and Msn4) in most cells.

**Image analysis and single-cell quantifications**. All image analysis was performed with custom Matlab (Mathworks) codes. First, bright-field images were segmented by circular Hough transformation (*CircularHough_GrdEx* function from Mathworks File Exchange) frame by frame to obtain single-cell masks. The masks were then tracked with custom codes based on u-track[65]. Finally, the tracked masks were applied to fluorescent images to acquire single-cell fluorescence signals. The results of segmentation and tracking were manually examined through custom GUI showing single-cell fluorescence traces and single-cell images of all channels in all frames. Cells with tracking errors were discarded. Cells at the corners of the imaging field were also discarded.

For the quantification of nuclear localization of TFs fused with fluorescent proteins, the difference between mean intensity of the five brightest pixels and median intensity of all pixels within a cell was used. To reduce the influence by background fluctuations, we first identified the frames that have positive nuclear localization signals if the mean distance between the top 10 brightest pixels was less than a given threshold. Next, the single-cell trace excluding these frames was fitted with cubic polynomial, and the fitting result was regarded as the background. For the quantification of maximum nuclear localization in response to stress (Supplementary Fig. 7e–g), the increment between maximum nuclear localization within 20 min after stress and median nuclear localization within 20 min before stress was calculated.

For the quantification of nascent transcriptional activity from PCP fluorescence, maximum z-projection was performed and the difference between mean intensity of the five brightest pixels and median intensity of all pixels within a cell was calculated.

For the quantification of nuclear signals of DRAQ7, the difference between mean intensity of the five brightest pixels and the median intensity of all pixels within a cell was calculated. The time of death for single cells was manually identified with a customized GUI (i.e., when the signal starts to increase sharply; as in Fig. 3c and Supplementary Movie 2).

For microscopy experiments without cell tracking (Supplementary Figs. 6e and 8e), nuclear localizations of Msn2 and Msn4 were determined based on the difference between the mean intensity of the five brightest pixels and median intensity of all pixels within a cell, as well as the mean distance between the top 10 brightest pixels. Cell death was identified if the nuclear signal of DRAQ7 raised above a given threshold.

**Pulse quantifications and pulse-triggered averaging analysis**. Pulses were identified by finding a local maximum whose amplitude exceeded an empirically defined threshold. In addition, the mean distance between the top 10 brightest pixels in the corresponding image should be smaller than an empirical threshold. During this process, shoulder-like peaks were also identified and were combined with neighboring peaks with higher amplitudes. The local maximum that satisfied the criteria was defined as the peak of a pulse. The data points on the two sides of the peak were fitted with splines separately and edges of half pulse were defined as the half-maxima of the fitted splines. The distance between the two edges was defined as pulse width and the integral of the fitted splines between the two edges was defined as pulse area (see Supplementary Fig. 1b for schematic).

For pulse classification, if the absolute time lag between Msn2 (or Msn2_cg) pulse and Msn4 (or Msn4_cg) pulse was not >2 min, these two pulses were classified as coincident; otherwise, the event was classified as noncoincident. For transcriptional response analysis (i.e., using PP7), Msn2- or Msn4-only pulse was classified based on whether there are pulses of the other TF within a 16-min window of one pulse.

For pulse-triggered averaging analysis of transcriptional responses, the maximum fluorescence intensity of the PCP foci within 8 min after an Msn2/Msn4 pulse minus the mean intensity within 8 min before the pulse was used as the transcriptional activity induced by the pulse.

**Autocorrelation and cross-correlation analysis**. Autocorrelation and cross-correlation were computed only for steady-state dynamics. To identify the steady-state response, the pulse within 8-min window of the transient response (i.e., when there are synchronous nuclear localization responses of Msn2 and Msn4) was identified as transient pulse and the median end time (i.e., the right edge of the pulse) of transient pulses was calculated. To reduce the influence of the transient response, twice the median end time was used as the start of the steady-state response. Autocorrelation of steady-state Msn2 (Msn4) dynamics and cross-correlation between steady-state Msn2 and Msn4 dynamics were computed with a maximum lag of 30 min using R function *acf* and *ccf*, respectively, for each cell.

Then, autocorrelation and cross-correlation of all cells were averaged and 95% confidence interval was computed by bootstrap (see below).

**Calculation of pulse coincidence rate**. To calculate pulse coincidence rate, only pulses during steady state were considered. To avoid the influence of pulse frequency on pulse coincidence rate, a method similar to bootstrap was adopted. Specifically, the same number of Msn2 (or Msn2_cg) pulses and Msn4 (or Msn4_cg) pulses were sampled with replacement under each stress level and the sample size was the same across all stress levels. The fraction of coincident pulses in the sampled pulses was calculated (the definition of coincident pulses was mentioned before). Sampling was repeated 1000 times and the average was computed as pulse coincidence rate, and 95% confidence interval was calculated as error bar.

We also computed the pulse coincidence rate without removing the effect of pulse frequency. Specifically, under each stress level, cell population was sampled to the same size with replacement and the fraction of Msn2 (or Msn2_cg) pulses that overlap with Msn4 (or Msn4_cg) pulses was calculated as the original pulse coincidence rate. Expected-by-chance fraction of coincidence was also computed as

$$\frac{\text{Number of Msn4 (or Msn4\_cg) pulses} \times \text{time window for overlapping identification}}{\text{total time}},$$

and was subtracted from the original pulse coincidence rate. Sampling was repeated 1000 times and the average was computed as pulse coincidence rate, and 95% confidence interval was calculated as error bar.

For the calculation of pulse coincidence rate at the single-cell level, the fraction of steady-state Msn2 (or Msn2_cg) and Msn4 (or Msn4_cg) pulses that are coincident pulses was calculated as single-cell pulse coincidence rate. For the classification of single cells as having high or low pulse coincidence rate between Msn2 and Msn4 (Supplementary Fig. 3d), cells were classified by comparing to the rate at the trough of density plot of pulse coincidence rate among single cells.

For dividing cells with similar total Msn2/4 pulse numbers into two subgroups based on the pulse coincidence rate in each cell (Fig. 3d and Supplementary Fig. 5d), we aimed to ensure that the two subgroups have similar cell numbers, and that cells with the same value of pulse coincidence rate should be included in the same subgroup. For most scenarios, we were able to obtain two similarly sized subgroups. However, because of the discrete nature of the single-cell pulse coincidence rate, two subgroups could have very different cell numbers (e.g., for pulse numbers 1–2).

For the calculation of nuclear colocalization percentage of Msn2 and Msn4 with snapshot data (Supplementary Fig. 6e), we needed to avoid the influence of nuclear localization fraction of individual TFs, as we have done so to avoid the influence of individual pulse frequency for the calculation of pulse coincidence rate. To do so, the same number of cells with nuclear-localized Msn2 and Msn4 were sampled with replacement for each time point and the sample size was the same across all time points. The fraction of cells with nuclear localization of both Msn2 and Msn4 in the sampled cells was calculated. Sampling was repeated 1000 times and the average was computed as nuclear colocalization percentage, and 95% confidence interval was calculated as error bar.

**Quantitative PCR and RNA-seq**. For steady-state experiments, overnight single-colony culture was diluted to $OD_{600} = 0.25$ with 10 mL media in a 50 mL centrifuge tube. After 4-h culture, cells from two 10 mL cultures were combined and briefly centrifuged. The supernatant was quickly discarded and cells were flash-frozen in liquid nitrogen before RNA extraction. For transient stress experiments, overnight single-colony culture was diluted to $OD_{600} = 0.5$ with 10 mL media without stress in a 50 mL centrifuge tube. After 4-h culture, 5 mL cell culture was mixed with 5 mL media containing double the designated stress concentration (1 M potassium chloride). The rest of the culture was mixed with 5 mL media without stress, which was used as the control group. Cells were then cultured by shaking under 30 °C for around 15 min and were harvested by brief centrifugation and flash-freezing.

RNA extraction was performed with HiPure HP Plant RNA Mini Kit (Magen) and RT-PCR was performed with iScript (Bio-Rad) or FastKing RT Kit (Tiangen). Typically, 1 μg RNA was used for RT-PCR per sample. Quantitative PCR was performed according to the standard protocol of GoTaq qPCR Master Mix (Promega) with 7500 FAST Real-Time PCR System (Applied Biosystems) and the results were analyzed with 7500 software v2.3 (Applied Biosystems). RQ value, which represented fold change using the delta-delta Ct method, was calculated and error bars represent 95% confidence intervals. *ACT1* was chosen as a reference gene.

For RNA-seq experiments, libraries were prepared according to standard Illumina protocols. Sequencing was performed on Illumina HiSeq X Ten. For transcriptional output measurements, all the libraries were sequenced in PE150 mode. Both library construction and sequencing were performed at GeneWiz. Sequencing reads were trimmed and filtered by Trimmomatic. The resulting sequences were aligned to yeast genome reference sacCer3 for *S. cerevisiae* (or CBS138 for *C. glabrata*) with Tophat. Alignments to gene loci were quantified with Cufflinks or featureCounts. For each gene, the fold change of fragments per kilobase per million mapped reads (or transcripts per million) between the stress group and the non-stress group was calculated. Differential gene expression analysis was performed with R package *DESeq2*[66]. Msn2/4 target genes (Fig. 2c and Supplementary Fig. 4c) were extracted from YEASTRACT[67], whose expression fold

change between the stress group (0.5 M KCl) and the control group (no KCl) in wild type (*S. cerevisiae*) should also be >1.2. Msn2/4_cg target genes (Fig. 5d and Supplementary Fig. 12e) were identified through differential gene expression analysis between wild type (*C. glabrata*) and *msn2_cg msn4_cg* under osmotic stress (0.5 M KCl, see Supplementary Fig. 12d).

For classification of redundant versus nonredundant targets of Msn2 and Msn4 (Fig. 3b and Supplementary Fig. 4e), Msn2/4 target gene was classified as redundant if the sum of its fold changes in *msn2* and *msn4* was higher than that in wild type and as nonredundant otherwise. Gene ontology analysis was performed with R package *clusterProfiler*[68].

**Flow cytometry**. For flow cytometry experiments, overnight single-colony culture was diluted to $OD_{600} = 0.005$. After 4-h culture with normal media, the fluorescent intensity of CFP was measured with a BD LSRFortessa flow cytometer in the Pacific Blue channel. BY4741 strain (wild type) was used as a negative control. Singlets were gated with EasyFlow (https://antebilab.github.io/easyflow).

**Mathematical model description**. A phenomenological model was constructed to understand the mechanism of temporal redundancy modulation. Specifically, PKA regulates both Msn2 and Msn4, while X mainly regulates Msn4. X was assumed to regulate Msn4 by phosphorylation as by PKA. In this model, stochastic and strong downregulation of PKA can induce Msn2/4 co-pulse, while weak downregulation can induce Msn2-only pulse. This is because PKA has a lower affinity for Msn2 (Supplementary Fig. 7). Downregulation of kinase X can induce Msn4-only pulse. This model was described based on Hill function by the following ordinary differential equations after parameter reduction through variable replacements:

$$\frac{d\text{Msn2}_n}{dt} = a_2 \times (\text{Msn2}_t - \text{Msn2}_n) \times \frac{1}{1 + \left(\text{PKA} \times \frac{K_d(\text{Msn4})}{K_d(\text{Msn2})}\right)^{n_2}} - b_2 \times \text{Msn2}_n \quad (1)$$

$$\frac{d\text{Msn4}_n}{dt} = a_4 \times (\text{Msn4}_t - \text{Msn4}_n) \times \frac{1}{1 + \text{PKA}^{n_4}} \times \frac{1}{1 + X^{n_X}} - b_4 \times \text{Msn4}_n \quad (2)$$

where PKA denoted concentration of PKA, X denoted the concentration of X, $\text{Msn2}_t$ ($\text{Msn4}_t$) denoted the total concentration of Msn2 (Msn4), $\text{Msn2}_n$ ($\text{Msn4}_n$) denoted the nuclear concentration of Msn2 (Msn4), $a_2$ ($a_4$) denoted the basal rate of nuclear importing of Msn2 (Msn4) without regulation by PKA (and X), $b_2$ ($b_4$) denoted the rate of nuclear exporting of Msn2 (Msn4), $K_d(\text{Msn2})$ ($K_d(\text{Msn4})$) denoted the dissociation constant for PKA regulating Msn2 (Msn4), $n_2$ ($n_4$) denoted the Hill coefficient for PKA regulating Msn2 (Msn4) and $n_X$ denoted the Hill coefficient for X regulating Msn4.

For stochastic simulations of Msn2 and Msn4 dynamics, a customized pipeline based on "τ-leap" method[69] was adopted. First, dynamics of PKA and X were assumed to exhibit periodic fluctuations that contain three different scenarios as illustrated in Supplementary Fig. 9c, i.e., upregulation, strong downregulation, or weak downregulation. Note that it has been widely accepted that dynamic PKA activity is a key mechanism driving Msn2/4 pulsatility, and we simply assumed that kinase X could exhibit activity dynamics similar to PKA. In addition, the choice of periodicity is just for the sake of simplicity and does not influence the results since the conclusion is unaffected if shuffling the pulses of PKA and X. The amplitudes for upregulation, strong downregulation, and weak downregulation were randomly chosen from predefined ranges, i.e., from 0 to 10% of the basal activity level, from 0 to 100% of the basal activity level, and from 0 to a predefined value (parameter $\text{amp}_{\text{weak}}$), respectively. Next, using the generated dynamics of PKA and X as inputs, the dynamics of Msn2 and Msn4 were simulated by "τ-leap" method according to the above equations (i.e., Eqs. (1) and (2)). Finally, the identification and quantification of Msn2 and Msn4 pulses and the calculation of pulse coincidence rate were performed as previously described. More specifically, the following parameter values were used for simulations: $a_2 = 1$, $a_4 = 1200$, $n_2 = n_4 = n_X = 4$, $b_2 = b_4 = 2$, $\text{Msn2}_t = \text{Msn4}_t = 200$, $X = 6$, $\text{amp}_{\text{weak}} = 4$, and the duration of simulation was 300 and the step size was 0.001. Parameter values for PKA and $K_d(\text{Msn4})/K_d(\text{Msn2})$ are shown in the figure legends.

**Proteome-wide analysis of pulsatile dynamics of TFs in *S. cerevisiae***. Pulsatile dynamics were identified in a proteome-wide movie-based screen in *S. cerevisiae*[25]. Only TFs screened in that work were included in our analysis. Identification of TFs was based on YeTFaSCo[70] and genes with DNA-binding domains or binding motifs were recognized as TFs. Dubious TFs in YeTFaSCo were excluded. Classifications of genes as with or without paralogs were based on Yeast Gene Order Browser[62].

**Dynamic index and functional redundancy of human TFs**. Classification of human genes as with or without paralogs was based on results of Dandage and Landry[71] and genes identified to have heteromeric paralogs in their work were excluded from our analysis. Gene (mainly TF) dynamic index was calculated as cell-to-cell variability in chromatin accessibility of its binding motifs and the analysis was performed by Buenrostro et al.[57] (see also Supplementary Fig. 13c for schematic). Data of six human cell lines were analyzed, namely, H1ESC: H1 human

embryonic stem cells; K562: K562 chronic myelogenous leukemia cells; GM12878: GM12878 lymphoblastoid cells; TF1: TF-1 cells (human erythroblast); HL60: HL-60 cells (human promyeloblasts); and BJ: BJ fibroblasts (human foreskin fibroblasts).

Functional redundancy of human genes with or without paralogs was obtained from CERES-estimated gene-knockout effect data from CRISPR-Cas9 essentiality screens in 342 cancer cell lines[72]. Score of 0 represents the median effect of nonessential genes (which indicates high functional redundancy) and −1 represents that of common core essential genes (which indicates low functional redundancy).

**Bootstrap and statistical analysis.** Bootstrap analyses were performed with the aforementioned sampling methods. One thousand resamplings were conducted and the mean and the 95% confidence interval of these 1000 results were used as the population average and error bar. If sampling methods were not mentioned, then a typical sampling method with replacement to the same size as the original was performed 1000 times.

For statistical analysis, Student's $t$ test was conducted with R function $t.test$, Kolmogorov–Smirnov test was performed with R function $ks.test$, Wilcoxon signed-rank test was performed with R function $wilcox.test$, and Fisher's exact test was performed with R function $fisher.test$. For linear regression analysis (Supplementary Fig. 5b), cells were binned by two dimensions (namely, total pulse number and coincident pulse number) and bins with ≥20 cells were used for analysis. Two linear regression models were constructed to explain the survival fraction based on total pulse number and coincident pulse number, or just total pulse number, respectively. Adjusted $R$-squared, sign of coefficients, and $p$ values of coefficients (two-sided $t$ test) were calculated with R function $lm$.

**Reporting summary.** Further information on research design is available in the Nature Research Reporting Summary linked to this article.

## Data availability

All sequencing data generated in this study have been deposited in publicly accessible databases. The processed and raw RNA-seq data are available in the NCBI Gene Expression Omnibus (GEO) database under the accession code GSE161373. The data that support the findings of this study are available from the corresponding author upon reasonable request. Source data are provided with this paper.

## Code availability

Custom scripts are available at https://github.com/wyzfcb/code_of_yeast_cell_fate_control_paper [73].

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

## Acknowledgements
We thank C. Tang and P. Wei for providing *S. pombe* and *C. glabrata* strains, respectively, C. Luo for help on microfluidics, and A. Moses and I. Hsu for comments on the manuscript. We also thank the imaging facility at the Center for Quantitative Biology at Peking University for equipment support. This work was supported by grants from National Natural Science Foundation of China (Grant No. 31771425) and National Key R&D Program of China (Grant Nos. 2018YFA0900703 and 2020YFA0906900).

## Author contributions
Y.W. and Y.L. conceived and designed the study. Y.W. performed the experiments. Y.W. and J.W. analyzed the data. M.D. and Y.L. contributed ideas for data analysis. Y.W. and Y.L. wrote the paper.

## Competing interests
The authors declare no competing interests.
