## [Peer Review File · Nature Communications]

REVIEWER COMMENTS

Reviewer #1 (Remarks to the Author):

In this manuscript, Wu et al revealed the temporal divergence in nuclear localization pulses of Msn2 and Msn4, two paralogous transcription factors (TFs) with largely redundant gene activation functions. They showed that this temporal divergence arises from different affinities of Msn2 and Msn4 to the upstream kinase and functions to enhance cell survival under stress. They further explored the relationship between the temporal divergence/pulsatile dynamics and the preservation of the paralog pairs through evolution by comparing their dynamics in multiple yeast species.

I find this paper interesting in that (1) it reveals a new mechanism underlying the different pulsatile dynamics of Msn2 and Msn4; (2) it demonstrates the functional relevance of TF pulses; (3) the interplay between dynamic regulation and paralog preservation is intriguing. Therefore, I recommend the acceptance of this manuscript for publication in Nature Communications if the authors are able to address the major points below.

Major concerns:

1. Fig. 3d shows that the temporal divergence can enhance cell survival under stress, which is a major finding of the paper. However, analysis of more cells will be needed. Their conclusion was primarily drawn from the bottom row (6 pulses): from left to right, the survival rate decreases with the number of coincident pulses. But, based on the color bars, each square in the bottom row only has ~20 cells. The survival rate with 0 coincident pulse (leftmost) is about 20%, so ~4 cells survived. The survival rate with 2 coincident pulses (rightmost) is ~15%, so 3 cells survived. It seems the authors based their major conclusion on this very small difference in the numbers of surviving cells. I would request much more cells to be included in this analysis.

2. Related to point 1, Fig. 3d only zoomed in on a small fraction of short-lived cells with few pulses. Will the conclusion still be valid if long-lived cells with more pulses (e.g. rows 7-20) are also included in the analysis?

3. Extended data Fig.6, a and b, shows that Msn2 and Msn4 have different affinities/sensitivities to PKA, which is another major finding of the paper. But an experiment that directly compares the Msn2 protein level in Strain 1 with Msn4 protein level in Strain 2 is missing. Although they are under the same Msn2 promoter, their protein degradation rates may be different, resulting in different protein levels and different localization levels (panel b). Given the importance of this result, I will suggest a quantitative immunoblotting experiment will be needed to exclude this possibility.

Reviewer #2 (Remarks to the Author):

In this manuscript, the authors use a combination of single-cell microscopy, transcriptomic studies, and mathematical modeling to study the dynamics of Msn2/Msn4, two transcription factor paralogs that regulate the general stress response in budding yeast. The authors first demonstrate that both Msn2 Msn4 localize to the nucleus in a pulse-like manner in response to external stress, and that the pulse coincidence rate of Msn2/Msn4 is modulated by the nature and strength of the applied stress. Next, the authors show that genetic regulation by the Msn2 and Msn4 paralogs is largely redundant, and that cells survive better when their pulse coincidence rate is lower. The authors then develop a mathematical model to confirm their hypothesis that the Msn2/Msn4 pulse coincidence rate is modulated by a second kinase "X" that only regulates the activity of Msn4. Finally, the authors investigate the potential evolutionary role of temporal redundancy modulation by evaluating how the activity of Msn2/Msn4 orthologs is modulated by stress in *S. pombe* (a pre-WGD relative that does not

display pulsing dynamics), *K. lactis* (a pre-WGD relative that displays pulsing dynamics) and *C. glabrata* (a post-WGD relative that displays pulsing dynamics).

Overall, the paper is well written and the experimental methods and sources of error are thoroughly discussed. While the manuscript contains a lot of data, one major point limiting my enthusiasm for this paper is that not all presented data helps push the main point of the paper across. Additionally, several of the arguments presented are flawed (see discussion below). It would be difficult for me to support this manuscript for publication in its current state unless issues associated with a significant portion of the results presented are addressed. I will briefly describe some of my concerns below.

First, as described by AkhavanAghdam (<https://elifesciences.org/articles/18458>), Msn2/Msn4 co-regulate downstream genes according to an "OR" relationships when genes can be turned ON quickly, and according to an "AND" relationship for slower genes. It is unclear whether the authors make that distinction in their analysis because the definition for redundant vs. non-redundant for genes does not explicitly take the "AND" genes into account. While not all genes operate as AND gates, those that do should not contribute to any fitness advantage and may even be misregulated when the pulse coincidence rate is low.

Second, a core claim of this manuscript is that "Temporal redundancy modulation provides fitness advantages under stress". However, all of the analysis supporting this claim is based on figures 3d and S5c and the evidence they provide is not very convincing. Why is the survival data not showing what happens for >6 Msn2/Msn4 pulses in Fig. 3c? Why are there so many cells with <6 total number of pulses when the raw data presented in Fig. S5a seem to show that most cells have several Msn2/Msn4 pulses? Furthermore, Fig. S5a shows that cells that die early still have some coincident pulses, and some of the cells surviving until the very end do not have any coincident pulse after time T=6h? Also, the authors claim in S5b that "a model including the temporal relationship between Msn2 and Msn4 pulses can explain the variation in cell survival much better than a model without", but they do not include any details about what the model is or does.

Third, the mathematical model presented in Fig. 4 is not very useful in understanding the proposed dynamics: yes, incorporating an additional kinase may reproduce some of the observed decrease in pulse coincidence rate, but many other models could also achieve the same behavior. For instance, what if Msn2 and Msn4 negatively interact with one another? Or what if each protein can only reside on the nucleus at a fixed stress-dependent density? What if another kinase is activating Msn2 and/or Msn4 instead of repressing Msn4? Additionally, since there is no coupling between Msn2 and Msn4 dynamics, the levels of Msn2/Msn4 are simply down-regulated by PKA/Kinase X levels. The model is thus highly dependent on the implied pulsatile behavior of PKA and kinase X to generate the Msn2/Msn4 pulses, which means that coincident pulses will only occur when PKA and kinase X pulse at the same time (which is what we see in Fig. 7b). Although the authors do acknowledge that this assumption may not be valid ("Note that the choice of periodic fluctuations is for the sake of simplicity in simulation and may not reflect the actual dynamics."), this all seems very artificial to me: it shifts the source of pulsatility away from Msn2/Msn4 to PKA/X without explaining how or why PKA/X levels themselves are pulsating.

Finally, I fail to see how the authors can claim that kinase X is Yak1: the data shown in Fig. 6e is far from convincing (especially since it relies on frequency measurements that are very close to zero). The authors should either i) cite relevant references justifying why they chose Yak1 as a candidate for kinase X and present more convincing evidence that Yak1 affects Msn4 only, ii) find a more convincing candidate for kinase X that does interact with Msn4, or iii) drop the hypothesis that a kinase X exists and instead use mathematical modeling in a different context --eg. to perform simulations that can make quantitative predictions about the link between temporal redundancy modulation and fitness.

The following minor points should be addressed:

- Fig. 5b/c: put a label on the figure to specify that this data is for *C. glabrata*.
- Fig. 5b: The authors use this figure to claim that "... both *S. cerevisiae* and *C. glabrata* preserved the temporal dynamics of Msn2 and Msn4 as well as their functional redundancy." Yet, from S9a+c, it seems like a rather large fraction of all pulses are not coincident in *C. glabrata*, and the pulsing dynamics seem to be mostly uncorrelated. The evidence does not support that claim, can the authors provide an explanation for this?
- Fig. S2a: label the x axis (how many hours of data?).
- Fig. S2b: there is useful information in the calculated auto-correlation and cross-correlation (CC) rates: the value of $CC(T=0)$ tells you about the fraction of peaks that are coincident, the fitting of an exponentially decaying function to $C(T>0)$ should give you the average width of the peaks, the presence of multiple exponentially decaying functions in the $C(T>0)$ fit indicates distinct pulsing dynamics, etc. Could this provide a more reliable and transparent measure for the pulse coincident rate?
- Fig. S5d/e: while the absolute frequency is a predictor of survival ability, its effect may be conflated with the pulse width and/or area of the peak. For instance, wouldn't the overall amount of time that Msn2/Msn4 spend inside the nucleus (ie. frequency times average area) be a better way to estimate its activity?
- Figs S1c and S8e: Hydrogen peroxide stress seems to behave in a qualitatively different manner compared to the other stresses at steady state (ie. very weak frequency modulation by stress level, very long duration/area at high stress for Msn2). This is OK in and of itself, but why did the authors use hydrogen peroxide stress to establish that Hsr1 does not pulsate in *S. pombe*? Would it make more sense to test pulsatile behavior under low glucose, high [KCl], or ethanol stress?
- Fig. S9c: why the shift from presenting the average pulse coincidence rate in Fig. 1c and S3 to presenting the proportion of cells with a low/high pulse coincidence rate in S9c? It seems like the high/low ratio provides a better insight into the biphasic nature of the pulsatile behavior, and it should be presented for all conditions for *S. cerevisiae* as well.

Reviewer #3 (Remarks to the Author):

In this paper, Wu et al. investigate the relationship between the dynamics of the transcription factors Msn2 and Msn4 in budding yeast. They show that following different stresses, pulses of Msn2 and Msn4 exhibit a relatively high degree of coincidence, which decreases as the stress increases. The authors use molecular biology, cell biology, transcriptomics and modeling to investigate the relationship between coincidence rate and stress response, and investigate a possible mechanism for modulation of the coincidence. Lastly, the authors extend their findings to other experimental systems, and use their findings to propose an evolutionary role for the potential advantages of coincidence modulation.

The model presented by the authors is unique and fundamentally important. However, the paper includes strong statements that many times are not supported by the presented data. In addition, alternatives to suggested mechanisms should be presented, discussed and tested experimentally before accepting the main model presented here.

Major comments

Fig 2c and S4b. The authors claim that since most genes appear in the center of the ternary plot, transcription is saturated for Msn2 and Msn4 alone. This is an important point since a core feature of the model relies on the transcriptional activity of Msn2/4 individually to be saturated under these conditions. However, this is not the only possibility and alternative models should be discussed. For

example, it is possible that when both Msn2 and Msn4 are present they regulate each other in a way that allows one to govern the early response and the other to govern late responses.

This point is central to the premise of the paper. One way to address it is to present a dose response curve for the transcriptional activity of the mutants. There is, in fact no evidence presented here that the transcriptional response to KCl is not digital but tunable, which raises the possibility that Msn2 and Msn4 act as binary regulators.

Furthermore, sampling genes only at one time point early after stress is introduced may miss important diverging dynamics. For example, it is possible that while many genes show early redundancy, late phase activation might look differently. It is also possible that the 15% of genes that don't show overlap between the two TFs, do overlap but are activated slower by one of the TFs. All these scenarios should be discussed and examined.

Fig. 3b and S4c The authors show that redundant genes exhibit higher activity (or a higher dynamic range) and immediately claim that temporal redundancy is modulated to allow for a greater dynamic range. There are several issues with this argument:

1. So far, the authors have not demonstrated temporal modulation, only raised this as an idea. Fig. 2 shows that pulses of Msn2 and Msn4 are redundant but not that this redundancy is modulated. It would be best to first present the phenomena and only later explain its potential function.
2. Fig. 3b shows that more TFs driving transcription leads to a higher dynamic range. Even when considering Fig 2 that shows that Msn2 and Msn4 are redundant for driving bursts of transcription (as opposed to accumulated transcript measured in 3b and S4c) the authors' explanation is not the only possible explanation, or even the simplest one. A competing hypothesis is that only one of the activators binds the promoter at any given time, and therefore when both are present the chances to bind increases (basically doubling the ON to OFF ratio over time). Moreover, the redundant targets might have longer mRNA half-lives which will contribute to their increase. A more thorough analysis that can help distinguish between these and other potential explanations is needed.
3. It is also worth noting that the response to glucose does not seem to saturate under these conditions for either the redundant or non-redundant genes. In order to prove the importance of modulation of coincidence, the comparison should be done under a dose for which the activity is saturated for each TFs individually. This would assure that increase in transcription can only be achieved by modulation of the coincidence rate (or at least eliminate one of the competing hypotheses).
4. The authors should display error bars as standard deviation and not standard error. The standard error would have applied if the error bars displayed biological replicates (how likely is the mean to fall within a certain range) but when looking at the distribution of fold changes between many different genes the standard deviation (showing the distribution of fold changes) is more suitable.

Fig. 3d, S5 The authors show that the coincidence rate negatively correlates with survivability. However according to the model every pulse should act as an activating pulse. A helpful way to display the data in 3d and S5c is comparing cells by their "effective pulses". That is, in the allotted time (7 hours?) how many unique pulses did a cell experience (so a coinciding pulse would be counted as one). While this can be derived from the heatmap in figure 3d and S5c, this type of presentation would be easier to understand and will allow direct comparisons between the WT and the mutants (extended 5d-e).

Furthermore, when looking at 3d I'm a bit lost as to why, if the model is correct, cells with 5 pulses in which 2 coincide, have lower survival than cells with 3 pulses with 0 coincidence (same number of unique pulses). Theoretically they have more chances to activate transcription even though they only effectively experience 3 pulses (according to the model). The way the data is displayed now makes it look as if the relationship holds only when looking at cells with equal frequencies and not across

frequencies, and if so it should be discussed.

Lastly, the analysis in 3d and 5c is limited due to the cutoff at 6 pulses. The small number of cells in each condition can be solved by binning cells into groups which will also strengthen the statistics (look at cells with 1-2 pulse, 3-5 pulses, 6-8 etc.). If there is a specific reason for the cutoff it should be explained.

S5c The authors claim that the correlation between coincidence and survival is dependent on transcription, as mutants of Msn2 and 4 that are unable to bind DNA don't show this correlation. However, based on Figure S5c it seems that cells lacking Msn2 and Msn4 survive better (50% survival vs 20% with the WT genes) and that the correlation is inverted (more coincidence leading to more survival). I might have misunderstood this analysis, but at the moment I find it confusing and contradictory to the main conclusions of the paper.

Fig S5d and e The authors show that in cells expressing either Msn2 or Msn4 increased pulse frequency correlates to increased survival. Other models should be considered here, including integrated Msn level, or duration of Msn level above a certain threshold as determinant of survival. In other words, does it matter whether low frequency is achieved by having sharp, narrow peaks of activity with long periods of 0 activity, vs. having wide broad peaks of activity with minimal time at 0 activity?

Fig. S6 The results presented in this figure are problematic. The assay does not prove that Msn4 has higher affinity for PKA, but only that Msn4 is less active when driven under the same promoter as Msn2. It is unknown, for example, if the levels of these two constructs under these conditions is similar. This needs to be shown experimentally by western blots or imaging. In addition, since the affinity argument is central to the model, an in vitro binding assay is needed and an in vitro kinase assay would be helpful too.

Fig 4 and S7 The authors propose factor X as an inhibitor of Msn4, which already has higher affinity to PKA. There are other alternative models that can explain the data and should be discussed. For example, the increased affinity of PKA to Msn4 might be dependent on X. In this model, a decrease in X will reduce the affinity of PKA to Msn4 below the affinity to Msn2, allowing for an Msn4 pulse in ranges that would still be inhibitory to Msn2.

To make the model meaningful the authors should clarify whether the parameters were derived from empirical measurements or were picked for best fit. For example, are the Msn2/ 4 activation thresholds based on real data? The figure legend states Msn2 and Msn4 binding affinities in a ratio of 0.5—what data supports that? A sensitivity/ robustness analysis is needed for the chosen parameters.

Fig S7h In order to show that the coincidence rate drops in the Msn2 mutants, the authors need to demonstrate that the pulsatile dynamics of Msn2 are unaltered. One prediction could be that the coincidence drops because these mutants are inactive, or do not exhibit a pulsatile dynamic anymore. Without these additional controls it is difficult to support the authors' claims.

Page 11 "Thus, ancestral Msn2/Msn4 acquired pulsing before WGD, consistent with our hypothesis that a mechanism for modulating their temporal relationship was evolved post duplication to preserve their functional redundancy."

Since *K. lactis* and *S. pombe* have only one Msn gene (which pulses in *K. lactis*), it is unclear how this proves that a mechanism for modulating Msn2/ 4 temporal relationship evolved post-genome duplication. It couldn't evolve preduplication, since there was only one Msn gene so there was no "relationship" to modulate. The authors have only proven that pulsing evolved pre-WGD. Moreover, the authors don't know WHY it evolved, they just claim that it supports fitness thus contributing to preservation. This statement needs to be adjusted.

Minor comments

Introduction:

"While the transient response dynamics can encode information about the inputs, the sustained dynamics after the transient phase enable complex signal processing mechanisms". There is very little explanation of the different phases of the dynamical response in the introduction. A more detailed explanation is needed especially for less systems-oriented readers.

Page 6: "It is interesting to note that, by altering temporal relationship between functionally redundant Msn2 and Msn4 pulses, cells can dynamically modulate the degree of redundancy between Msn2 and Msn4, resulting in an effective modulation of redundancy in a temporal manner (i.e., temporal redundancy modulation)." At this point of the paper this statement should be presented as an hypothesis.

"We next analyzed how cells could benefit from the modulation of temporal redundancy." This should also be changed since modulation of redundancy has not been demonstrated.

Page 9 "We identified a candidate kinase Yak1" It will be helpful to elaborate how this kinase was identified – was it through a screen? Literature search? An educated guess? Were there are other potential candidates?

Dear Reviewers,

We greatly appreciate your time and the constructive comments and suggestions, which have truly helped us to improve the manuscript. We are glad that you acknowledged the potential significance of our work. In response to your comments and suggestions, we have performed a series of new experiments and analyses with complementary techniques, and have obtained additional evidences supporting the function and mechanism of temporal redundancy modulation of Msn2 and Msn4. We believe that, with these new results, our work has provided important insights into temporal dynamics in genetic circuits.

In the revised manuscript, we have incorporated these new results, and have largely improved the text and figures to comprehensively address all your comments. These are described in the point-by-point reply. Here we first summarize the major changes in the revision:

- We have provided new data and analyses supporting the applicability of our model for cells with more than 6 pulses. More specifically, we conducted additional single-cell stress survival experiments and collected more cells. In addition to performing analyses analogous to the ones in the original manuscript, we have included new analyses that further substantiate our claim.
- We have thoroughly discussed alternative hypotheses raised by reviewers #2 and #3 in the revision. These discussions are included in the main text as well as a new figure panel (**Supplementary Fig. 8a**).
- We have carried out new experiments testing the differential affinities to PKA for Msn2 and Msn4. The results agree with our original finding and provide key insights into the mechanism of temporal redundancy modulation.
- Additionally, we have provided new analyses and/or experiments supporting several key claims in our manuscript, including the saturation of target responses by individual Msn2/4 pulses, Yak1 as the candidate kinase regulating Msn4, the difference between pulse frequency and other pulse characteristics, and the robustness of model parameters.

In the revised text, changes are highlighted in red. Below please find a summary of changes to the figures:

Main figures:

1. **Fig. 2b**: a minor error in the original code was corrected and the plot was updated, which does not affect the main claim of the data.
2. **Fig. 3b**: replaced the original plot (with error bars) with violin plots showing the distribution of data points together with p values from Kolmogorov-Smirnov tests.
3. **Fig. 3d**: new experimental data was included and the data was analyzed by binning cells with similar pulse numbers. This new plot shows the results for all pulse numbers (including >6 pulses).

4. **Fig. 3e:** a new figure panel showing the relationship between survival ability and effective/total pulse number in wild-type cells.
5. **Fig. 5b:** added a label of *C. glabrata*.
6. **Fig. 5d:** added a label of *C. glabrata*.
7. **Fig. 5e:** removed the traces of functional redundancy and dynamics divergence before WGD to improve accuracy in the cartoon.

Supplementary figures:

1. **Supplementary Fig. 2a:** added the label of x axis.
2. **Supplementary Fig. 2c:** cross-correlation in **Supplementary Fig. 2b** at time lag zero.
3. **Supplementary Fig. 2d:** a new figure panel showing cross-correlation at time lag zero for cells with comparable nuclear localization intensities across glucose limitation stress levels.
4. **Supplementary Fig. 4a:** same modifications as **Fig. 2b**.
5. **Supplementary Fig. 4b:** a new figure panel showing average trajectories of Msn2 and Msn4 in wild-type cells during the response to a sudden salt stress.
6. **Supplementary Fig. 4d:** a new figure panel showing the saturation of Msn2 and Msn4 in regulating 4xSTRE and DDR2 at the single pulse level.
7. **Supplementary Fig. 4e:** same modifications as **Fig. 3b**.
8. **Supplementary Fig. 5a:** reduced the font sizes of the dots. These randomly sampled 400 cells were from the new dataset (>1000 cells).
9. **Supplementary Fig. 5b:** updated the original analysis with the new dataset.
10. **Supplementary Fig. 5c:** a new figure panel showing fold differences of survival fraction increment between adding an extra effective pulse versus adding an extra total pulse.
11. **Supplementary Fig. 5d:** updated analysis for DBD mutant data with the new method (analogous to **Fig. 3d**).
12. **Supplementary Fig. 5e:** a new figure panel showing the relationship between survival ability and effective/total pulse number in DBD mutant strain.
13. **Supplementary Fig. 5h:** a new figure panel showing the average pulse area, duration, amplitude, and frequency times area for groups with different survival abilities in *msn2* or *msn4* mutant strain.
14. **Supplementary Fig. 5i:** a new figure panel showing the decrease of pulse coincidence rate along the time course under glucose limitation stress.
15. **Supplementary Fig. 5j:** a new figure panel showing the changes of Msn2/4 nuclear co-localization percentages over time under low or high glucose condition.
16. **Supplementary Fig. 6b:** a new figure panel showing the temporal responses of Msn2 in strain 1 and Msn4 in strain 2 to sudden PKA inhibition.
17. **Supplementary Fig. 6c:** a new figure panel showing the flow cytometry quantifications of Msn2-CFP in strain 1 and Msn4-CFP in strain 2.
18. **Supplementary Fig. 6d:** a new figure panel showing the fluorescence microscopy quantifications of Msn2-CFP in strain 1 and Msn4-CFP in strain 2.
19. **Supplementary Fig. 6e:** added the average trajectories of Msn2-CFP in strain 1 and Msn4-CFP in strain 2 next to the original plot.
20. **Supplementary Fig. 6f:** a new figure panel showing the maximum nuclear localization responses and the average trajectories of Msn2-CFP in strain 1 and Msn4-CFP in strain 2

to ethanol stress.

21. **Supplementary Fig. 6g:** a new figure panel showing the maximum nuclear localization responses and the average trajectories of Msn2-CFP in strain 1 and Msn4-CFP in strain 2 to glucose limitation stress.
22. **Supplementary Fig. 7b:** a new figure panel showing the flowchart of Yak1 identification.
23. **Supplementary Fig. 7e:** a new figure panel showing the percentage of cells with nuclear localization of Msn2 and Msn4 under different induction levels of Yak1.
24. **Supplementary Fig. 8a:** a new figure panel for discussing about alternative models.
25. **Supplementary Fig. 8f-g:** new figure panels showing the sensitivity analysis of the model parameters.
26. **Supplementary Fig. 11a:** replaced cells #3 and #4 with two other cells that have coincident pulses to demonstrate that there are also coincident pulses in *C. glabrata*.
27. **Supplementary Fig. 11c:** replaced the original pie charts with the distributions of single-cell pulse coincidence rates to ensure consistency with **Fig. 1** and **Supplementary Fig. 3**.

Point-by-point reply (reviewer comment in blue and author response in black):

Reviewer #1 (Remarks to the Author):

In this manuscript, Wu et al revealed the temporal divergence in nuclear localization pulses of Msn2 and Msn4, two paralogous transcription factors (TFs) with largely redundant gene activation functions. They showed that this temporal divergence arises from different affinities of Msn2 and Msn4 to the upstream kinase and functions to enhance cell survival under stress. They further explored the relationship between the temporal divergence/pulsatile dynamics and the preservation of the paralog pairs through evolution by comparing their dynamics in multiple yeast species.

I find this paper interesting in that (1) it reveals a new mechanism underlying the different pulsatile dynamics of Msn2 and Msn4; (2) it demonstrates the functional relevance of TF pulses; (3) the interplay between dynamic regulation and paralog preservation is intriguing. Therefore, I recommend the acceptance of this manuscript for publication in Nature Communications if the authors are able to address the major points below.

We greatly appreciate the very helpful comments and suggestions by the reviewer. We have incorporated several suggestions by the reviewer in our revision, which have allowed us to substantiate the role and mechanism of pulse coincidence modulation. Below we provide a detailed response to each concern.

Major concerns:

1. Fig. 3d shows that the temporal divergence can enhance cell survival under stress, which is a major finding of the paper. However, analysis of more cells will be needed. Their conclusion was primarily drawn from the bottom row (6 pulses): from left to right, the survival rate decreases with the number of coincident pulses. But, based on the color bars, each square in the bottom row only has ~20 cells. The survival rate with 0 coincident pulse (leftmost) is about 20%, so ~4 cells survived. The survival rate with 2 coincident pulses (rightmost) is ~15%, so 3 cells survived. It seems the authors based their major conclusion on this very small difference in the numbers of surviving cells. I would request much more cells to be included in this analysis.

We agree with the reviewer that the single-cell survival result would benefit from a larger dataset. And because this data is essential for testing our hypothesized model, we have designed and carried out a series of new experiments and analyses to address this important issue. In the revised manuscript, we have included these new results, which now provide multiple lines of evidence supporting our model.

In the following responses, we first reiterate the concern raised by the reviewer and discuss the rationales and limitations of the original **Fig. 3d**, and then explain how our new results provide much stronger supports for our model.

- 1) In the original manuscript, to test our model that Msn2/4 pulse redundancy could be modulated to regulate stress survival, we compared the survival data of wild-type versus the DBD mutant strain, and resorted to multiple analysis approaches. These analysis approaches include the comparison of survival fraction for cells of the same total pulses but different coincident pulses (**Fig. 3d**), as well as the comparison of linear regression models that include different independent variables (original **Extended Data Fig. 5b**). We agree with the reviewer that **Fig. 3d** is relatively qualitative and suffers from the relatively small numbers of cells. We address this issue by acquiring much more cells and by implementing new analyses. More importantly, we designed and carried out a new experiment to further test our model.
- 2) To address the issue with **Fig. 3d**, we carried out new additional survival assay experiment, and have now obtained more than 1000 cells that were monitored over 13 hours of movie at 2-minute frame interval. With this new dataset, we performed four complementary analyses to test our model and model predictions.
 - 1) In the first analysis, we tested that for cells with the same total Msn2/4 pulse number, whether the ones with low pulse coincidence rates would have a higher stress survival capability. This analysis was originally carried out in **Fig. 3d**, but as pointed out by the reviewer, each subset contained a relatively small number of cells and it was challenging to arrive at a statistically significant conclusion. To enhance the statistical power, we binned cells having similar pulse numbers (e.g., 1-2, 3-4, 5-6 pulses), and obtained two sub-groups of cells for each binned group of cells based on the pulse coincidence rate in each cell (see **Methods** for details). By doing so, we increased the number of cells for each sub-group and thus enhancing the statistical power, and we could also test our model for cells with more pulses (larger than 6 pulses). We found

that for six different groups of cells (containing varying pulse numbers), the subgroups with lower pulse coincidence rates all have higher stress survival rates compared to their counterparts. Additionally, cells with more total pulse numbers have higher stress survival rates. And importantly, these relationships are abolished in the DBD mutant strain. These results provide a strong support for our model (see **Fig. 3d** and **Supplementary Fig. 5d, Line 256-264, 296-301** in the text).

- 2) In the second analysis, we aimed to test our model in a more quantitative manner by constructing linear regression models, which was originally carried out in the original **Extended Data Fig. 5b**. With the new updated dataset, we binned cells based on the total pulse number and the number of coincident pulses, and repeated the analysis. We found that the linear model including the number of coincident pulses can better explain the survival capabilities of different cell groups, providing an additional support for our model (see **Supplementary Fig. 5b, Line 266-275** in the text).
 - 3) In the third analysis, we aimed to test our model prediction that, with the same number of effective versus total Msn2/4 pulses, cells should possess different survival capabilities. This is because that coincident Msn2/4 pulses would be counted as one effective pulse since Msn2/Msn4 pulses are largely functionally redundant. More specifically, our model predicts that cells with a certain number of effective pulses would have higher survival capabilities compared to cells with the same number of total pulses. To test this, we calculated the stress survival fractions for groups of cells with different effective pulse numbers and with corresponding total pulse numbers. We found that with the same number of total pulse or effective pulse, the latter group of cells has a higher stress survival capacity for almost all the pulse numbers analyzed. This result offers a key line of evidence supporting our model (see **Fig. 3e, Line 277-286** in the text).
 - 4) Analogous to the above prediction, our model further predicts that adding an extra effective pulse would yield a larger increase in cell survival capability compared to adding an extra total pulse. This is because an extra total pulse could coincide with existing pulses (and thus would not help the cell), while an extra effective pulse (by definition) would lead to additional expression of stress response genes. To test this, we computed the fold-change between the cell survival contributions by a single effective pulse and by a single total pulse. We found that under most scenarios, adding an extra effective pulse indeed results in a larger contribution to cell survival compared to adding an extra total pulse. This result further substantiates our model (see **Supplementary Fig. 5c, Line 288-294** in the text).
 - 5) Together, using our new dataset, these four different analyses provide strong supports for our model that budding yeast cells appear to modulate pulse coincidence rate between Msn2 and Msn4 pulses to regulate stress survival.
- 3) To further strengthen our conclusion, we carried out a new experiment that includes at least 5000 cells per time point to test our model prediction. More specifically, as stress level increases, our model predicts that budding yeast cells would separate the coincident Msn2/4 pulses (i.e., reduce pulse coincidence rate) to increase the expression levels of stress response genes. In our original stress survival assay at 0.05% glucose, we observed that the fraction of coincident pulses appears to decrease as the culture condition worsens (as time

increases), which is consistent with our model prediction (see **Supplementary Fig. 5i**). To strengthen the support for our model, we thus modified the experiment: instead of tracing individual cells by taking fast time points, we took one frame per hour and captured more than 5000 cells per time point, allowing us to accurately quantify the pulse coincidence rate of surviving cells at each time point. With this dataset, we found that indeed the pulse coincidence rate of the cell population decreases over time, consistent with the picture that cells modulate the pulse coincidence rate to enhance stress survival (see **Supplementary Fig. 5j, Line 318-329** in the text).

Together, we believe these new data and analyses have provided much stronger supports for our conclusion and have greatly improved the manuscript.

2. Related to point 1, Fig. 3d only zoomed in on a small fraction of short-lived cells with few pulses. Will the conclusion still be valid if long-lived cells with more pulses (e.g. rows 7-20) are also included in the analysis?

We thank the reviewer for raising this concern, which was similarly raised by other reviewers. We agree that it is critical to test whether our model also holds for cells with larger pulse numbers, which could not be tested with our original dataset due to the relatively small number of cells. We address this issue with our new data by performing multiple analyses for cells with larger than 6 total pulses. More specifically,

- 1) In the new analysis analogous to the original **Fig. 3d**, as mentioned in the above response, we binned cells having similar pulse numbers (e.g., 1-2, 3-4, 5-6 pulses), and obtained two sub-groups of cells for each binned group of cells based on the pulse coincidence rate in each cell (see **Methods** for details). In this analysis, we included groups of cells with larger than 6 pulses (i.e., 7-8, 9-12, and >12), and found that for these cells that have relatively large pulse numbers, the sub-groups with lower pulse coincidence rates also exhibit higher stress survival capabilities, similar to the cells with smaller pulse numbers (≤ 6). Therefore, this new analysis suggests that even for cells that have larger pulse numbers, cells appear to modulate the pulse coincidence rate to enhance stress survival (see **Fig. 3d, Line 256-264** in the text).
- 2) Our newly added analysis also supports the applicability of our model for cells with higher pulse numbers. More specifically, by comparing the stress survival capabilities of cells with the same total pulse number versus the same effective pulse number, we found that with the same number of effective versus total Msn2/4 pulses, the former group of cells has a higher survival capability for almost all pulse numbers. Importantly, this relationship holds for cells with high pulse numbers as well as for cells with low pulse numbers, suggesting that our model holds for cells with both low and high pulse numbers (see **Fig. 3e, Line 277-286** in the text).

Together, these new results strongly suggest that our model not only applies for cells with lower pulse numbers, but also applies for cells with higher pulse numbers. We have incorporated these

new results in the revision, which help to further strengthen our conclusion.

3. Extended data Fig.6, a and b, shows that Msn2 and Msn4 have different affinities/sensitivities to PKA, which is another major finding of the paper. But an experiment that directly compares the Msn2 protein level in Strain 1 with Msn4 protein level in Strain 2 is missing. Although they are under the same Msn2 promoter, their protein degradation rates may be different, resulting in different protein levels and different localization levels (panel b). Given the importance of this result, I will suggest a quantitative immunoblotting experiment will be needed to exclude this possibility.

We thank the reviewer for pointing out this important control experiment. We agree with the reviewer that there could be post-transcriptional mechanisms that contribute to differential Msn2 and Msn4 protein expression levels even under the same promoter. More generally, in order to strengthen the support for our proposed mechanism of pulse redundancy modulation, we need to address two key questions related to the reviewer's comment:

- a) Could the difference in nuclear localization level arise from the difference in Msn2/4 protein expression levels, rather than from the differences in PKA affinities?
- b) Are there additional evidences supporting the differential PKA affinities for Msn2/4?

To address these questions, we performed a series of new experiments and have obtained new evidences supporting our model that different Msn2/4 nuclear localization levels likely arise from their differential PKA affinities rather than the difference in their protein expression levels. We next describe the details of these new experiments.

- 1) To address the first question, we incorporated the suggestion by the reviewer and attempted to quantify the protein levels of Msn2 and Msn4 using different techniques. More specifically, we aimed to test the hypothesis that for the two mutant strains shown in **Supplementary Fig. 6a**, the reduced nuclear localization level of Msn4 arises from the reduced Msn4 protein expression level compared to Msn2, rather than from Msn4's higher affinity to PKA. A key premise of this hypothesis is that the protein expression level of Msn4 is lower than that of Msn2.
 - a) To test this, we quantified cellular Msn2 and Msn4 protein levels in these two mutant strains using both flow cytometry and fluorescence microscopy, as both proteins are fused with CFP. Flow cytometry data showed that the CFP signal distribution for Msn4-CFP is slightly right shifted compared to that for Msn2-CFP, indicating that there are more Msn4 protein molecules (see **Supplementary Fig. 6c, Line 357-358** in the text). In the second assay, cellular CFP quantifications using fluorescence microscopy showed that mean Msn4-CFP level per cell is ~7.4% higher than Msn2-CFP (see **Supplementary Fig. 6d, Line 357-358** in the text).
 - b) Results from both techniques show that the mean Msn4 protein level is slightly higher than Msn2, which cannot explain the observed differences in their nuclear localization levels (i.e., $Msn4 < Msn2$). In other words, the differential localization levels are likely due to the differential affinities to PKA, as we originally hypothesized.

- 2) To address the second question, we aimed to collect more data to support our hypothesis that the different nuclear localization levels are due to the differential affinities to PKA. In the original manuscript, we only had one piece of evidence supporting this hypothesis, which should be much strengthened as pointed out by reviewer #3. We carried out three different assays to address this issue.
- a) In the first assay, we aimed to compare the nuclear localization dynamics of Msn2 and Msn4 in response to sudden inhibition of PKA activity. Our model predicts that if Msn2/4 possess differential affinities to PKA, then their nuclear localization responses to PKA inhibition would be different. More specifically, the TF with a lower affinity to PKA would be dephosphorylated faster, resulting in a more rapid shuttling into the nucleus. Using the two mutant strains illustrated in **Supplementary Fig. 6a**, we measured the nuclear localization responses of initially cytoplasmic Msn2-CFP and Msn4-CFP upon the addition of H-89, a validated PKA inhibitor in budding yeast. The data showed that the PKA inhibitor H-89 induced a much more rapid cytoplasm-to-nucleus shuttling of Msn2-CFP compared to Msn4-CFP. In other words, this data suggests that under the same inhibited PKA activity, Msn2 is much more easily dephosphorylated (and shuttled into the nucleus) compared to Msn4, providing a strong support for our model that Msn2 has a lower affinity to PKA than Msn4 (see **Supplementary Fig. 6b, Line 347-356** in the text).
 - b) In the second assay, we further compared the nuclear localization dynamics of Msn2-CFP and Msn4-CFP under glucose limitation stress. During glucose limitation, PKA is first inhibited and then re-activated. When examining the nuclear localization responses of Msn2/4, we observed a clear difference in the dynamics of the response, i.e., while both TFs enter the nucleus with the same timescale, Msn4 displays a much faster timescale of exit from the nucleus compared to Msn2. This data suggests that Msn4 is re-phosphorylated by PKA at a faster time scale, which is consistent with the picture that Msn4 has a higher affinity to PKA compared to Msn2 (see **Supplementary Fig. 6g, Line 358-364** in the text).
 - c) In the third assay, we further compared their nuclear localization dynamics under both KCl and ethanol stresses. Both stresses induce fast nuclear localizations of Msn2 and Msn4, but the transient pulses of Msn4-CFP had a significantly lower amplitude compared to that of Msn2-CFP for both stresses. This result is consistent with the picture that Msn4 is more phosphorylated by PKA and thus its nuclear localization is weaker compared to Msn2. This data thus further substantiates our hypothesis (see **Supplementary Fig. 6e-f, Line 358-364** in the text).
 - d) New evidences from these three assays together provide strong supports for our model that Msn2 and Msn4 have differential affinities to PKA and their nuclear localization dynamics are differentially regulated by PKA.

Together, we believe that these results further substantiate our model and have greatly strengthened our manuscript.

Reviewer #2 (Remarks to the Author):

In this manuscript, the authors use a combination of single-cell microscopy, transcriptomic studies, and mathematical modeling to study the dynamics of Msn2/Msn4, two transcription factor paralogs that regulate the general stress response in budding yeast. The authors first demonstrate that both Msn2 Msn4 localize to the nucleus in a pulse-like manner in response to external stress, and that the pulse coincidence rate of Msn2/Msn4 is modulated by the nature and strength of the applied stress. Next, the authors show that genetic regulation by the Msn2 and Msn4 paralogs is largely redundant, and that cells survive better when their pulse coincidence rate is lower. The authors then develop a mathematical model to confirm their hypothesis that the Msn2/Msn4 pulse coincidence rate is modulated by a second kinase "X" that only regulates the activity of Msn4. Finally, the authors investigate the potential evolutionary role of temporal redundancy modulation by evaluating how the activity of Msn2/Msn4 orthologs is modulated by stress in *S. pombe* (a pre-WGD relative that does not display pulsing dynamics), *K. lactis* (a pre-WGD relative that displays pulsing dynamics) and *C. glabrata* (a post-WGD relative that displays pulsing dynamics).

Overall, the paper is well written and the experimental methods and sources of error are thoroughly discussed. While the manuscript contains a lot of data, one major point limiting my enthusiasm for this paper is that not all presented data helps push the main point of the paper across. Additionally, several of the arguments presented are flawed (see discussion below). It would be difficult for me to support this manuscript for publication in its current state unless issues associated with a significant portion of the results presented are addressed. I will briefly describe some of my concerns below.

We greatly appreciate the very helpful comments and suggestions by the reviewer. We have incorporated several suggestions by the reviewer in our revision, which have allowed us to substantiate the role and mechanism of pulse coincidence modulation. Below we provide a detailed response to each concern.

First, as described by AkhavanAghdam (<https://elifesciences.org/articles/18458>), Msn2/Msn4 co-regulate downstream genes according to an "OR" relationships when genes can be turned ON quickly, and according to an "AND" relationship for slower genes. It is unclear whether the authors make that distinction in their analysis because the definition for redundant vs. non-redundant for genes does not explicitly take the "AND" genes into account. While not all genes operate as AND gates, those that do should not contribute to any fitness advantage and may even be misregulated when the pulse coincidence rate is low.

We thank the reviewer for raising this point. We agree that it is important to clarify the similarities and differences between the two approaches for classifying target genes, i.e., AND/OR-gate targets and redundant/non-redundant targets. In the following response, we explain how genes are classified as AND/OR targets or redundant/non-redundant targets, and discuss whether our analysis distinguishes between AND and OR genes.

First, technical differences between these two classification methods lead to different metrics for classifying genes. As mentioned by the reviewer, in the work by AkhavanAghdam and co-authors, the authors adopted the classification scheme in their earlier work^{1,2} where genes were classified based on the speed of turning on transcription. And the speed of gene turning on was quantified under artificially activated Msn2, allowing the successful identification of genes with different kinetic parameters governing promoter activation. In our work, the definition of redundant versus non-redundant targets was accomplished by comparing the fold-changes of target expression at the mRNA level in three different strains, namely *msn2*, *msn4* and wild type, in response to a transient stress. Therefore, while the former method classifies genes based on kinetic parameters of target activation, the latter method classifies genes based on the regulator dependence of target expression level.

Because of the difference in the classification methods (as well as the difference in strain background), we did not attempt to compare between different definitions of gene groups. Furthermore, it remains to be determined whether AND or OR genes are enriched for specific functions. In the revised manuscript, we noted that it would be intriguing to further investigate how redundant/non-redundant genes could be different in activation kinetics in order to compare with AND/OR-based classification (**Line 187-189**).

To further address this issue, we performed a gene ontology analysis to examine the significantly enriched functions of redundant versus non-redundant genes. We found that the top three enriched functions of redundant genes are “response to abiotic stimulus” ($p = 5.1e-7$), “cellular response to chemical stimulus” ($p = 1.1e-4$), and “response to osmotic stress” ($p = 2.7e-4$). In contrast, the top three enriched functions of non-redundant genes are “regulation of transcription by RNA polymerase II” ($p = 1.1e-5$), “negative regulation of transcription by RNA polymerase II” ($p = 1.5e-4$), and “positive regulation of transcription by RNA polymerase II” ($p = 1.5e-3$). The sharp distinction between the enriched functions for redundant versus non-redundant genes is consistent with the important role of the modulation of the temporal relationship between Msn2 and Msn4 (see **Line 180-187** in the text).

Second, a core claim of this manuscript is that "Temporal redundancy modulation provides fitness advantages under stress". However, all of the analysis supporting this claim is based on figures 3d and S5c and the evidence they provide is not very convincing. Why is the survival data not showing what happens for >6 Msn2/Msn4 pulses in Fig. 3c?

The reviewer raised two related concerns: a) the evidence for cell survival regulation by pulse redundancy modulation is not sufficient; b) it is unclear whether our model holds for cells with higher pulse numbers. To address these two important issues, we have designed and carried out a series of new experiments and analyses. In the revised manuscript, we have included these new results, which now provide multiple lines of evidence supporting our model for cells with varying pulse numbers. More specifically,

To address both concerns, we carried out new additional survival assay experiments, and have now obtained more than 1000 cells that were monitored over 13 hours of movie at 2-minute frame interval. With this new dataset, we performed four complementary analyses to test our model and model predictions.

- 1) In the first analysis, we tested that for cells with the same total Msn2/4 pulse number, whether the ones with low pulse coincidence rates would have a higher stress survival capability. This analysis was originally carried out in **Fig. 3d**, but as pointed out by the first reviewer, each subset contained a relatively small number of cells and it was challenging to arrive at a statistically significant conclusion. To enhance the statistical power, we binned cells having similar pulse numbers (e.g., 1-2, 3-4, 5-6 pulses), and obtained two sub-groups of cells for each binned group of cells based on the pulse coincidence rate in each cell (see **Methods** for details). By doing so, we increased the number of cells for each sub-group and thus enhancing statistical power, and we could also test our model for cells with more pulses (larger than 6 pulses). More specifically, we included groups of cells with larger than 6 pulses (i.e., 7-8, 9-12, and >12), and found that for these cells that have relatively large pulse numbers, the sub-groups with lower pulse coincidence rates also exhibit higher stress survival capabilities, similar to the cells with smaller pulse numbers (≤ 6). Additionally, cells with more total pulse numbers have higher stress survival rates. And importantly, these relationships are abolished in the DBD mutant strain. Therefore, these results provide a strong support for our model, and even for cells that have larger pulse numbers, cells appear to modulate the pulse coincidence rate to enhance stress survival (see **Fig. 3d** and **Supplementary Fig. 5d, Line 256-264, 296-301** in the text).
- 2) In the second analysis, we aimed to test our model in a more quantitative manner by constructing linear regression models, which was originally carried out in the original **Extended Data Fig. 5b**. With the new updated dataset, we binned cells based on the total pulse number and the number of coincident pulses, and repeated the analysis. We found that the linear model including the number of coincident pulses can better explain the survival capabilities of different cell groups, providing an additional support for our model (see **Supplementary Fig. 5b, Line 266-275** in the text).
- 3) In the third analysis, we aimed to test our model prediction that, with the same number of effective (note that each coincident pulse is counted as one effective pulse in our model) versus total Msn2/4 pulses, cells should possess different survival capabilities. This is because that there are coincident Msn2/4 pulses, and Msn2/Msn4 pulses are functionally redundant. More specifically, our model predicts that cells with a certain number of effective pulses would have higher survival capabilities compared to cells with the same number of total pulses. To test this, we calculated the stress survival fractions for groups of cells with different effective pulse numbers and with corresponding total pulse numbers. We found that with the same number of total pulse or effective pulse, the latter group of cells has a higher stress survival capacity for almost all the pulse numbers analyzed. This result offers a key line of evidence supporting our model (see **Fig. 3e, Line 277-286** in the text).
- 4) Analogous to the above prediction, our model further predicts that adding an extra effective pulse would yield a larger increase in cell survival capability compared to adding an extra

total pulse. This is because an extra total pulse could coincide with existing pulses (and thus would not help the cell), while an extra effective pulse (by definition) would lead to additional expression of stress response genes. To test this, we computed the fold-change between the cell survival contributions by a single effective pulse and by a single total pulse. We found that under most scenarios, adding an extra effective pulse indeed results in a larger contribution to cell survival compared to adding an extra total pulse. This result further substantiates our model (see **Supplementary Fig. 5c, Line 288-294** in the text).

Thus, using our new dataset, these four different analyses provide strong supports for our model that budding yeast cells appear to modulate pulse coincidence rate between Msn2 and Msn4 pulses to regulate stress survival. And these new results suggest that our model not only applies for cells with lower pulse numbers, but also applies for cells with higher pulse numbers.

To further strengthen our conclusion, we carried out a new experiment that includes at least 5000 cells per time point to test our model prediction. More specifically, as stress level increases, our model predicts that budding yeast cells would separate the coincident Msn2/4 pulses (i.e., reduce pulse coincidence rate) to increase the expression levels of stress response genes. In our original stress survival assay at 0.05% glucose, we observed that the fraction of coincident pulses appears to decrease as the culture condition worsens (as time increases), which is consistent with our model prediction (see **Supplementary Fig. 5i**). To strengthen the support for our model, we thus modified the experiment: instead of tracing individual cells by taking fast time points, we took one frame per hour and captured more than 5000 cells per time point, allowing us to accurately quantify the pulse coincidence rate of surviving cells at each time point. With this dataset, we found that indeed the pulse coincidence rate of the cell population decreases over time, consistent with the picture that cells modulate the pulse coincidence rate to enhance stress survival (see **Supplementary Fig. 5j, Line 318-329** in the text).

Together, we believe all these new results have provided significant evidences supporting our main conclusion. These results have been incorporated into the revised manuscript, which help to greatly improve the manuscript.

Why are there so many cells with <6 total number of pulses when the raw data presented in Fig. S5a seem to show that most cells have several Msn2/Msn4 pulses?

We thank the reviewer for raising this issue. The sizes of the dots in the original **Extended Data Fig. 5a** were enlarged for easier visualization because the original sizes were too small. Because of such a visual effect, each dot may appear to belong to different cells, causing the confusion that many cells have more pulses than their real pulse numbers.

We apologize for this confusion and we have now reduced the sizes of the dots. Additionally, we have also provided a histogram showing the number of cells versus total pulse number (**Fig. R1**), which should allow an accurate representation of the data.

Figure R1. Histogram of total pulse number before death for all cells in survival experiments of Fig. 3d.

Furthermore, Fig. S5a shows that cells that die early still have some coincident pulses, and some of the cells surviving until the very end do not have any coincident pulse after time $T=6h$?

We thank the reviewer for pointing out this intriguing phenomenon. There is indeed a decrease in the pulse coincidence rate at later time points, but this trend could be over-exaggerated because the sizes of the dots were enlarged in the original plot. To investigate this phenomenon, we quantified how pulse coincidence rate changes along the time course (**Supplementary Fig. 5i**). Intriguingly, the result shows that cells appeared to modulate the pulse coincidence rate as they stayed longer in the glucose limitation environment. This observation is consistent with a scenario that the glucose concentration dropped at later time points, effectively increasing the stress level for the cells. Under this scenario, cells should indeed reduce the pulse coincidence rate as our model suggests.

To further study this intriguing observation, we carried out new experiments with much larger cell numbers under both low (0.05%) and high (2%) glucose concentrations. In these new experiments, instead of tracing individual cells by taking fast time points, we took one frame per hour and captured more than 3000 cells per time point, allowing us to accurately quantify the pulse coincidence rate of surviving cells at each time point. Results from this assay show that cells appeared to lower their pulse coincidence rate at later time points only when they are under low glucose condition (**Supplementary Fig. 5j**), consistent with the scenario that the stress level increases as glucose level decreases.

Together, we believe these results provide further supports for our model that yeast cells could modulate the pulse coincidence rate to enhance stress survival (see **Line 318-329** in the text).

Also, the authors claim in S5b that "a model including the temporal relationship between Msn2

and Msn4 pulses can explain the variation in cell survival much better than a model without", but they do not include any details about what the model is or does.

We apologize for the lack of a detailed description of the regression models. In the following, we will provide more details regarding these models and describe how this issue is addressed in the revision.

In order to quantitatively analyze the role of pulse coincidence, we resorted to two linear regression models (partial or full model) to explain the cell survival fraction by either one or two independent variables, where the partial model uses only the total pulse number and the full model uses both the total pulse number and the coincident pulse number. By fitting our data with these two models, we showed that the full model has a higher explanatory power than the partial model, as shown by the adjusted R-squared (which had been adjusted for the number of variables in the models). Thus, we concluded that a model including the temporal relationship between Msn2 and Msn4 pulses (namely, including coincident pulse number) can explain the variation in cell survival better than a model without. Note that the same conclusion also holds when comparing these two models with our new survival dataset.

In the revised manuscript, we have clarified this issue by revising the figure legend of **Supplementary Fig. 5b**, and by including the details on the regression models in both the main text and the **Methods** sections (see **Line 266-275 and 1043-1048**).

Third, the mathematical model presented in Fig. 4 is not very useful in understanding the proposed dynamics: yes, incorporating an additional kinase may reproduce some of the observed decrease in pulse coincidence rate, but many other models could also achieve the same behavior. For instance, what if Msn2 and Msn4 negatively interact with one another?

We thank the reviewer for raising an important issue regarding alternative models for pulse coincidence modulation. We apologize for not providing related information in the original manuscript. Based on the comments by both reviewer #2 and reviewer #3, we have now compiled a list of alternative models that could enable pulse coincidence modulation as well as a list of experimental observations that the model should be consistent with (see **Supplementary Fig. 8a**). For each potential model, we determined whether each of the experimental observations is consistent with the model. By doing so, we found that the model we proposed originally is so far the only one that could be consistent with all four experimental observations. However, in the revised manuscript, we emphasized that it is likely that we have missed some important mechanisms, and our model is only meant to offer explanations for the current experimental observations (**Line 408-414**). Thus, while our model may not be the only model that explains our single-cell data, it sheds a light into how the divergent temporal dynamics of paralogous TFs could be achieved. In the following, we will specifically discuss each of the alternative models raised by the reviewer.

Regarding the model where "Msn2 and Msn4 negatively interact with one another", this is

indeed an intriguing model where Msn2 could negatively affect the nuclear localization of Msn4 at higher stress levels to reduce pulse coincidence rate. However, this model does not appear to be consistent with multiple experimental observations as discussed below.

- 1) The model would require direct or indirect mechanisms for one protein to affect the nuclear localization of the other protein. We first looked for the evidence supporting direct protein-protein interactions between these two proteins. Based on the existing literatures and the BioGRID database, while there are genetic interactions between them, we could not identify any physical protein-protein interactions between Msn2 and Msn4 (see **Fig. R2**).

MSN2 *Saccharomyces cerevisiae* (S288c)
stress-responsive transcriptional activator MSN2, L000001198, YMR037C

Stress-responsive transcriptional activator; activated in stochastic pulses of nuclear localization in response to various stress conditions; binds DNA at stress response elements of responsive genes; relative distribution to nucleus increases upon DNA replication stress

Switch View: **Interactors (180)** Interactions (290) Network PTM Sites (15)

Displaying 180 total unique interactors
Sort By: [Evidence] [Alphabetical]

MSN4 | YKL062W, L000001199 34 [details]
Stress-responsive transcriptional activator; activated in stochastic pulses of nuclear localization in response to various stress conditions; binds DNA at stress response elements of responsive genes, inducing gene expression; involved in diauxic shift

PHO UBI

Experimental Evidence Code	Role	Dataset	Throughput	Curated By	Notes
Dosage Rescue	HIT	Schmitt AP (1996)	Low	BioGRID	
Negative Genetic	HIT	Bandyopadhyay S (2010)	High	BioGRID	
Phenotypic Enhancement	HIT	Versele M (2004)	Low	BioGRID	
	HIT	Fabrizio P (2004)	Low	BioGRID	
	HIT	Hasan R (2002)	Low	BioGRID	
	HIT	Moskvina E (1998)	Low	BioGRID	
	HIT	Boy-Marcotte E (1998)	Low	BioGRID	
	HIT	Inoue Y (1998)	Low	BioGRID	
	HIT	Martinez-Rastor MT (1996)	Low	BioGRID	
	HIT	Estruch F (1993)	Low	BioGRID	
	HIT	Zaehring H (2000)	Low	BioGRID	
	BAIT	Gatti X (2005)	Low	BioGRID	
	HIT	Medvedik O (2007)	Low	BioGRID	
	HIT	Watanabe D (2010)	Low	BioGRID	
	BAIT	Alejandro-Osorio AL (2009)	Low	BioGRID	
	BAIT	Sadeh A (2011)	Low	BioGRID	
	HIT	Welch AZ (2012)	Low	BioGRID	
HIT	Lee YJ (2013)	Low	BioGRID		
HIT	Li L (2015)	Low	BioGRID		
HIT	Li L (2017)	Low	BioGRID		
HIT	Kuang Z (2017)	Low	BioGRID		
HIT	Zemva J (2017)	Low	BioGRID		
HIT	Yi JK (2016)	Low	BioGRID		
HIT	Li S (2018)	Low	BioGRID		
HIT	Chapal M (2019)	Low	BioGRID		
Synthetic Growth Defect	HIT	Medvedik O (2007)	Low	BioGRID	
	HIT	Jimenez A (2010)	Low	BioGRID	
	HIT	Calahan D (2011)	Low	BioGRID	
	HIT	Hirsola MG (2014)	Low	BioGRID	
	HIT	Rajyashni PK (2017)	Low	BioGRID	
HIT	Yamaguchi Y (2018)	Low	BioGRID		
Synthetic Lethality	HIT	Schueler C (2004)	Low	BioGRID	
Synthetic Rescue	HIT	Hosiner D (2009)	Low	BioGRID	
	BAIT	Sadeh A (2011)	Low	BioGRID	

Figure R2. No evidence found for direct protein-protein interactions between Msn2 and Msn4 in the BioGRID database. The figure is a screenshot of the interactions between Msn2 and Msn4 in BioGRID database and there are only 34 genetic interactions in this database while no physical interactions are found.

- 2) We next speculated that there could be stress-level-dependent indirect mechanisms that enable one protein to push the other protein out of the nucleus, leading to low pulse coincidence rate at high stress levels. As an example, Msn2 and Msn4 could compete for limited resources such as importins that are necessary for nuclear import. However, this scenario is not consistent with our observations that under transient responses to high stress levels (e.g., KCl stress), both Msn2 and Msn4 are activated simultaneously and exhibit relatively high levels of nuclear localization (**Supplementary Fig. 4b**). In other words, if Msn2 and Msn4 could negatively impact each other's nuclear localization, it would be unlikely to observe simultaneous high levels of nuclear localization of both proteins. Thus,

this experimental data suggests that it is unlikely for Msn2 and Msn4 to negatively affect each other's nuclear localization in an indirect manner.

Together, although this mechanism proposed by the reviewer seems intriguing, it is not supported by existing experimental evidences. We have also clarified this point in the revision (see **Supplementary Fig. 8a**).

Or what if each protein can only reside on the nucleus at a fixed stress-dependent density?

To a certain extent, what was proposed here by the reviewer is a coarse-grained view of what goes on with Msn2 and Msn4 at the population level based on our experimental data. That said, the "stress-dependent density" would refer to our observation that each protein appears to reside in the nucleus for different fractions of time that are stress-level dependent. However, the exact mechanism that accounts for the fractional regulation of Msn2/4 nuclear localization is missing. With our model, we aimed to provide an explanation of how the coarse-grained view of Msn2/4 behaviors could be achieved by temporal activities of upstream kinases at the single-cell level.

What if another kinase is activating Msn2 and/or Msn4 instead of repressing Msn4?

If there is another kinase (instead of Yak1 as we proposed) which activates Msn4, then it could indeed explain several of our experimental observations (see **Supplementary Fig. 8a**). In this scenario, this kinase would need to induce Msn4 nuclear localization by increasing the phosphorylation level of the nuclear localization or export signal of Msn4. Yet we did not identify such a kinase in our screen. However, because we only performed a small screen (we now provided the flowchart explaining our screen, see **Supplementary Fig. 7b**), it is likely that we could have missed it in our screen. Therefore, based on our limited experimental results so far, we cannot rule out the existence of such a kinase proposed by the reviewer. We have now clearly discussed the limitations of our model and the possibility of alternative models (see **Supplementary Fig. 8a, Line 408-414**).

Additionally, since there is no coupling between Msn2 and Msn4 dynamics, the levels of Msn2/Msn4 are simply down-regulated by PKA/Kinase X levels. The model is thus highly dependent on the implied pulsatile behavior of PKA and kinase X to generate the Msn2/Msn4 pulses, which means that coincident pulses will only occur when PKA and kinase X pulse at the same time (which is what we see in Fig. 7b).

Yes - our model depends on the pulsatile behaviors of PKA and kinase X, the rationales for which will be explained in the response to the next comment.

Regarding the coupling between Msn2/Msn4 dynamics, their nuclear localization dynamics are known to be both affected by PKA. In this sense, their dynamics are actually coupled as

depicted in our model. And because of the two proteins' differential affinities to PKA, coincident pulses could occur when PKA pulses downwards with enough amplitude. This scenario was shown by green shading in **Supplementary Fig. 8c**.

Although the authors do acknowledge that this assumption may not be valid ("Note that the choice of periodic fluctuations is for the sake of simplicity in simulation and may not reflect the actual dynamics."), this all seems very artificial to me: it shifts the source of pulsatility away from Msn2/Msn4 to PKA/X without explaining how or why PKA/X levels themselves are pulsating.

We would like to first re-emphasize that our model is a phenomenological model instead of a full mechanistic model (which was explicitly stated in our original manuscript, see "Mathematical model description" section in **Methods**). Because of the phenomenological nature of the model, many details of the model were thus simplified or omitted. We will next address the two concerns raised here: a) whether it is reasonable to assume pulsatile activities of PKA and kinase X; and b) whether it is valid to simulate the model with periodic dynamics.

For the first concern, what the reviewer mentioned is very accurate – we "shifted the source of pulsatility away from Msn2/Msn4 to PKA/X". The rationale for making this assumption is that it has been widely accepted in the literatures that dynamic PKA activity is a key mechanism driving Msn2/4 pulsatility. This assumption/mechanism was clearly explained in our original manuscript. More specifically, PKA is well-known to be regulated by negative feedback, and was recently theoretically modeled in a related paper and the simulation results were compared to experimental data³. However, as for kinase X, we indeed do not have any evidence for its activity dynamics. And we simply assumed that it could exhibit activity dynamics similar to PKA. To clarify this, we have now discussed the assumptions and limitations of our model in the **Methods** section (**Line 994-996**).

For the second concern, we would like to clarify further that the periodic dynamics were only used to generate temporal patterns of kinase activity fluctuations in the simulations. The rationale for doing so is that this phenomenological model was only intended for capturing the features of our data in a qualitative manner, and we thus did not attempt to simulate the underlying molecular circuits for generating PKA/X pulsatility. And more importantly, the exact temporal patterns of the pulsatility should not affect the main conclusion of the model. For example, in **Supplementary Fig. 8c**, we could shuffle the pulses of PKA and X in time and the conclusion is unaffected. To clarify this, we have now added additional descriptions of the simulation in the **Methods** section (**Line 996-998**).

Finally, I fail to see how the authors can claim that kinase X is Yak1: the data shown in Fig. 6e is far from convincing (especially since it relies on frequency measurements that are very close to zero). The authors should either i) cite relevant references justifying why they chose Yak1 as a candidate for kinase X and present more convincing evidence that Yak1 affects Msn4 only, ii)

find a more convincing candidate for kinase X that does interact with Msn4, or iii) drop the hypothesis that a kinase X exists and instead use mathematical modeling in a different context --eg. to perform simulations that can make quantitative predictions about the link between temporal redundancy modulation and fitness.

We thank the reviewer for pointing out this issue. Based on the reviewer's suggestions, we have now provided a detailed flowchart explaining how we chose candidate kinases from existing literatures and databases, and how the screening experiments were carried out (see **Supplementary Fig. 7b**).

To strengthen our hypothesis that Yak1 is kinase X which represses Msn4 alone, we conducted a new experiment (containing many more cells than the original dataset) to compare Msn2/4 pulse frequency under different Yak1 levels. Briefly, we cultured the strain with copper-inducible CUP1 promoter for perturbing Yak1 level with different levels of Cu²⁺ and took images for bright-field, Msn2 (CFP) and Msn4 (YFP). Then single cells were segmented and nuclear localization of Msn2 and Msn4 was identified for each cell (see **Methods** for more details). As shown in **Supplementary Fig. 7e**, elevating Yak1 level only decreased the percentage of cells with nuclear localization of Msn4 while did not change that of Msn2, indicating that increasing Yak1 decreased Msn4 pulse frequency alone. This result, together with **Supplementary Fig. 7d**, further supports that Yak1 appears to be the kinase X repressing Msn4 alone in our experimental system. However, as we noted above, our screen is relatively limited and we have now clearly stated this limitation and have discussed the possibility of other kinases or phosphatases regulating the dynamics of Msn4 (**Line 408-410**).

The following minor points should be addressed:

-Fig. 5b/c: put a label on the figure to specify that this data is for *C. glabrata*.

We thank the reviewer for this suggestion and we have added a label on the figure in the revision.

-Fig. 5b: The authors use this figure to claim that "... both *S. cerevisiae* and *C. glabrata* preserved the temporal dynamics of Msn2 and Msn4 as well as their functional redundancy." Yet, from S9a+c, it seems like a rather large fraction of all pulses are not coincident in *C. glabrata*, and the pulsing dynamics seem to be mostly uncorrelated. The evidence does not support that claim, can the authors provide an explanation for this?

We thank the reviewer for pointing out these important issues that we should clarify. Our hypothesis is that paralogs could maintain functional redundancy by evolving divergent temporal dynamics. It is important to note that this hypothesis does not predict that the paralogs should retain a high pulse coincidence rate, but rather it predicts that cells could modulate the pulse coincidence rate in response to changes in environment. The results in **Fig. 5b** show that although the overall pulse coincidence rates are low for *C. glabrata*, cells are able to modulate

pulse coincidence rate when stress level changes, which are consistent with our model prediction.

We would like to further emphasize another important implication of the *C. glabrata* data. When comparing to *S. cerevisiae*, Msn2_cg and Msn4_cg in *C. glabrata* show relatively low pulse coincidence rates, which means that the two paralogs have more divergent pulsing dynamics in *C. glabrata* than in *S. cerevisiae*. Intriguingly, the RNA-seq results showed that Msn2/4 target genes in *C. glabrata* are more redundantly activated compared to *S. cerevisiae*. These results indicate that the more divergent the pulsing dynamics evolved, the higher degree of functional redundancy could be preserved for the paralogous TFs. In other words, it appears that dynamics divergence could be ‘traded’ with functional redundancy (in terms of target gene activation). This implication was clearly stated in the revised manuscript (**Line 483-487**) and was depicted in the summary cartoon of **Fig. 5e**.

-Fig. S2a: label the x axis (how many hours of data?).

We thank the reviewer for pointing out this error. The duration of single-cell trajectories was 4 hours and we have added it to the label of x axis of **Supplementary Fig. 2a**.

-Fig. S2b: there is useful information in the calculated auto-correlation and cross-correlation (CC) rates: the value of $CC(T=0)$ tells you about the fraction of peaks that are coincident, the fitting of an exponentially decaying function to $C(T>0)$ should give you the average width of the peaks, the presence of multiple exponentially decaying functions in the $C(T>0)$ fit indicates distinct pulsing dynamics, etc. Could this provide a more reliable and transparent measure for the pulse coincident rate?

We thank the reviewer for these helpful suggestions. We agree that cross-correlation at time lag zero is a reliable measure of pulse relationship between Msn2 and Msn4. We have plotted cross-correlation values at time lag zero with the raw trajectories under different conditions (**Supplementary Fig. 2c**). Additionally, as we have done for the analysis of pulse coincidence rate, we resampled each cell population to control for the changes in pulse dynamics under different conditions, and found that indeed the cross-correlation at time lag zero decreases as stress level increases (**Supplementary Fig. 2d**). This observation is consistent with the trend in pulse coincidence rate, providing further support for our claim that yeast cells can modulate pulse coincidence rate in response to stresses.

As for the decay in the cross-correlation function, we felt that it is rather complicated to obtain information regarding the pulse interaction, because their pulse durations as well as pulse frequencies change under different stress conditions (**Supplementary Fig. 1c**).

-Fig. S5d/e: while the absolute frequency is a predictor of survival ability, its effect may be

conflated with the pulse width and/or area of the peak. For instance, wouldn't the overall amount of time that Msn2/Msn4 spend inside the nucleus (ie. frequency times average area) be a better way to estimate its activity?

We thank the reviewer for raising this issue. To address this, we have computed pulse area, pulse duration, pulse amplitude, and pulse frequency multiplied by pulse area for the two populations with different stress survival abilities (**Supplementary Fig. 5f-h**). We found that pulse area, duration, or amplitude do not positively correlate with stress survival. Furthermore, although pulse frequency multiplied by pulse area positively correlates with stress survival, the p-value is not as significant as that of pulse frequency alone. To strengthen this point, we made another figure showing that increasing pulse area does not lead to increased transcriptional activities for both DDR2 and 4xSTRE synthetic gene., indicating that the areas of most Msn2/4 pulses have exceeded the saturation point of the target's dose-response curve (**Supplementary Fig. 4d**).

-Figs S1c and S8e: Hydrogen peroxide stress seems to behave in a qualitatively different manner compared to the other stresses at steady state (ie. very weak frequency modulation by stress level, very long duration/area at high stress for Msn2). This is OK in and of itself, but why did the authors use hydrogen peroxide stress to establish that Hsr1 does not pulsate in *S. pombe*? Would it make more sense to test pulsatile behavior under low glucose, high [KCl], or ethanol stress?

We apologize that we did not clearly clarify the background on Hsr1. Even though Hsr1 is Msn2/4 ortholog in *S. pombe*, it is not a regulator of general stress responses like Msn2/4 in *S. cerevisiae*. Instead, Hsr1 is a specific regulator for hydrogen peroxide stress⁴. Thus, we chose hydrogen peroxide to characterize its activation dynamics. We have further clarified this point in the revision (**Line 445-448**).

-Fig. S9c: why the shift from to presenting the average pulse coincidence rate in Fig. 1c and S3 to presenting the proportion of cells with a low/high pulse coincidence rate in S9c? It seems like the high/low ratio provides a better insight into the biphasic nature of the pulsatile behavior, and it should be presented for all conditions for *S. cerevisiae* as well.

We thank the reviewer for the helpful suggestions. We have carefully re-examined the metric shown in **Fig. S9c**, i.e., the fraction of cells with low/high pulse coincidence rate, and found that this metric is not always consistent with the mean pulse coincidence rate. This is because that the grouping of cells into low or high pulse coincidence rate depends on the position of the trough in the distribution, which however is not constant and shifts with stress condition. To better illustrate this issue, we performed simulations to examine the relationship between mean pulse coincidence rate and the fraction of cells with high pulse coincidence rate (**Fig. R3a**). We found that these two metrics are not necessarily correlated. We also compared these two metrics using the experimental data, and observed a similar inconsistency (**Fig. R3b**). Because of this

issue, we have decided to remove this representation of the data. We have thus replaced **Fig. S9c** with the distribution of pulse coincidence rate similar to **Fig. S3** (see **Supplementary Fig. 11c**). We believe that pulse coincidence distribution is a better way to present the data.

Reviewer #3 (Remarks to the Author):

In this paper, Wu et al. investigate the relationship between the dynamics of the transcription factors Msn2 and Msn4 in budding yeast. They show that following different stresses, pulses of Msn2 and Msn4 exhibit a relatively high degree of coincidence, which decreases as the stress increases. The authors use molecular biology, cell biology, transcriptomics and modeling to investigate the relationship between coincidence rate and stress response, and investigate a possible mechanism for modulation of the coincidence. Lastly, the authors extend their findings

to other experimental systems, and use their findings to propose an evolutionary role for the potential advantages of coincidence modulation.

The model presented by the authors is unique and fundamentally important. However, the paper includes strong statements that many times are not supported by the presented data. In addition, alternatives to suggested mechanisms should be presented, discussed and tested experimentally before accepting the main model presented here.

We greatly appreciate the very helpful comments and suggestions by the reviewer. We have incorporated several suggestions by the reviewer in our revision, which have allowed us to substantiate the role and mechanism of pulse coincidence modulation. Below we provide a detailed response to each concern.

Major comments

Fig 2c and S4b. The authors claim that since most genes appear in the center of the ternary plot, transcription is saturated for Msn2 and Msn4 alone. This is an important point since a core feature of the model relies on the transcriptional activity of Msn2/4 individually to be saturated under these conditions. However, this is not the only possibility and alternative models should be discussed. For example, it is possible that when both Msn2 and Msn4 are present they regulate each other in a way that allows one to govern the early response and the other to govern late responses.

This point is central to the premise of the paper. One way to address it is to present a dose response curve for the transcriptional activity of the mutants. There is, in fact no evidence presented here that the transcriptional response to KCl is not digital but tunable, which raises the possibility that Msn2 and Msn4 act as binary regulators.

We thank the reviewer for the helpful comments and suggestions. As mentioned by the reviewer, our explanation for the functional redundancy between Msn2 and Msn4 is that every pulse of Msn2 or Msn4 ‘hits’ the target response curve nearby the saturation region of the curve. In other words, we hypothesized that individual redundant target genes of Msn2 and Msn4 are on average activated to a similar level by every Msn2 or Msn4 pulse, regardless of the size of the pulse or the identity of the pulse (Msn2 or Msn4). An experimental demonstration of this assumption is thus important, not only for supporting our model, but also for ruling out alternative hypotheses such as the one mentioned by the reviewer (however, as we discussed in the response to reviewer #2’s comment on p14-16, we note that there is currently no experimental evidence supporting the physical interactions between Msn2 and Msn4).

To experimentally test our hypothesis, we resorted to the real-time transcriptional reporter assay in which we simultaneously monitored the nuclear localization dynamics of Msn2 and Msn4 together with downstream target gene transcription (**Fig. 2**). More specifically, we aimed to test whether every Msn2 or Msn4 pulse saturated the dose response curves of both the natural target

gene (DDR2-24xPP7) and the synthetic target gene (4xSTRE-24xPP7). To do so, we binned Msn2 or Msn4 pulses based on the area of each pulse (defined in **Supplementary Fig. 1b**), and used the pulse-triggered averaging approach to compute the corresponding transcriptional activities of either the natural or the synthetic target gene. By doing so, we found that both target genes' transcriptional activities are relatively comparable across different bins of Msn2 or Msn4 pulses (**Supplementary Fig. 4d**).

This result suggests that individual pulses of Msn2 or Msn4 likely saturate the dose response curves of both natural and synthetic targets. More importantly, this result further supports our hypothesis that the shared target genes of Msn2 and Msn4 can be redundantly activated by pulses of Msn2 or Msn4.

Furthermore, sampling genes only at one time point early after stress is introduced may miss important diverging dynamics. For example, it is possible that while many genes show early redundancy, late phase activation might look differently. It is also possible that the 15% of genes that don't show overlap between the two TFs, do overlap but are activated slower by one of the TFs. All these scenarios should be discussed and examined.

We thank the reviewer for raising a potential limitation in our screen for redundant versus non-redundant genes. In the following, we will first clarify the rationales and assumptions for our assay, provide experimental evidences supporting the assumptions, and discuss alternative scenarios mentioned by the reviewer.

First, we would like to clarify the rationales underlying the transient KCl stress assay for identifying redundant versus non-redundant genes. Ideally, to identify redundant vs. non-redundant targets, time-lapse imaging assay as in **Fig. 2a-b** should be implemented to quantify all Msn2/4 target gene responses, which however is extremely challenging. To circumvent this challenge and to quantify how all Msn2/4 targets respond to Msn2 pulse, Msn4 pulse, or co-pulse, we needed to 'synthesize' all three types of pulses in cell populations, which would allow us to quantify expression responses using bulk RNA-seq. To synthesize such pulses, we first created three different yeast strains containing either Msn2, or Msn4, or both. We then subjected these three strains to transient KCl stress, which should induce synchronous Msn2/4 pulses across individual cells in the population. And because the transient pulses end at around 15 min, cells were collected at this time point to analyze the transcriptome response to Msn2/4 pulses.

In this assay, two assumptions were made, which are supported by or are consistent with our experimental data. More specifically,

- 1) We assumed that in our experimental condition, transient KCl stress indeed can induce a synchronous pulse of Msn2 and Msn4, allowing us to quantify the transcriptome response at the 15 min time point. To verify this, we plotted the averaged Msn2 and Msn4 dynamics in response to a transient step increase in KCl and quantified the fraction of responsive cells. The results show that ~79% of cells induce Msn2/4 co-pulse, and the averaged Msn2/4 dynamics indeed exhibit a transient pulse that ends at around 20 min (**Supplementary Fig.**

- 4b).**
- 2) We also assumed that transient pulses of Msn2 or Msn4 should saturate most target response curves, allowing us to identify redundant versus non-redundant genes by comparing target responses to the transient Msn2-only, Msn4-only, and Msn2/4 co-pulse. This assumption is consistent with our observations: a) the transient pulses are relatively large and should fall in the saturation region of the dose response curves (**Supplementary Fig. 4b**); and b) we observed that most targets lie in the center of the ternary plot, highly indicative of saturated responses when Msn2 and Msn4 are both present (comparing to only Msn2 or only Msn4).

Based on these clarifications, we now address the concerns raised by the reviewer.

- 1) For the first concern on *missing important diverging dynamics by sampling at the early time point*, as we clarified above, the purpose of the assay is to capture the response to only the transient pulse (that is synchronized across cells), which requires us to sample at a relatively early time point. As for the diverging dynamics mentioned by the reviewer, we actually wanted to avoid such diverging and unsynchronized dynamics especially in the wild-type strain, where Msn2 and Msn4 are both synchronously activated in the transient pulse (and de-synchronize afterwards). To clarify this, we have now included additional explanations of the assay design in the revised text (**Line 160-166**).
- 2) For the second concern regarding *the alternative scenarios for redundant or non-redundant targets*, the scenarios suggested by the reviewer do not appear to be consistent with the experimental data shown for the 2nd assumption above. More specifically, our classification of redundant versus non-redundant genes is based on the responses of target genes to single pulses of Msn2 and/or Msn4. In other words, the fundamental premise of our model is that Msn2/4 are activated in pulses and we focus on the responses of targets to individual pulses of Msn2/4, rather than their responses to sustained activations of Msn2/4 (which was implicated in the alternative scenarios proposed by the reviewer). In the revised text, we have now clearly discussed this point (**Line 160-166**).

We believe that these changes have clarified the concerns and have much improved the accuracy of our statements.

Fig. 3b and S4c The authors show that redundant genes exhibit higher activity (or a higher dynamic range) and immediately claim that temporal redundancy is modulated to allow for a greater dynamic range. There are several issues with this argument:

1. So far, the authors have not demonstrated temporal modulation, only raised this as an idea. Fig. 2 shows that pulses of Msn2 and Msn4 are redundant but not that this redundancy is modulated. It would be best to first present the phenomena and only later explain its potential function.

We would like to first clarify the logic of our data and describe the changes we made to address the issue raised by the reviewer. We first demonstrated that cells appear to modulate pulse coincidence rate under different stress levels in both **Fig. 1** and **Fig. S3**, which is regarded as

the temporal modulation of Msn2/4 pulse relationship. We then moved on to demonstrate the functional redundancy between Msn2 and Msn4 pulses, where we showed that pulses of Msn2 and Msn4 can redundantly activate two target genes using PP7 assay (**Fig. 2a-b**), and identified more target genes that can be redundantly activated by pulses of Msn2 and Msn4 (**Fig. 2c**). Based on the evidences from **Fig. 1** and **Fig. 2**, we speculated that cells could modulate the Msn2/4 pulse relationship to regulate gene expression of redundant targets (schematic in **Fig. 3a**). We then tested this speculation by quantifying the gene expression fold change under a series of glucose conditions (which are the same as in **Fig. 1c**), and found that redundant targets display a larger dynamic range compared to the non-redundant targets (**Fig. 3b**). Together, results from **Fig. 1** to **Fig. 3b** led us to conclude that “temporal redundancy is modulated to allow for a greater dynamic range”.

We realized that it is indeed confusing to mention “temporal redundancy modulation” before **Fig. 3b**. We have now incorporated the suggestion by the reviewer and moved the terminology of “temporal redundancy modulation” after the introduction of the data in **Fig. 3b (Line 200-235)**.

2. Fig. 3b shows that more TFs driving transcription leads to a higher dynamic range. Even when considering Fig 2 that shows that Msn2 and Msn4 are redundant for driving bursts of transcription (as opposed to accumulated transcript measured in 3b and S4c) the authors’ explanation is not the only possible explanation, or even the simplest one. A competing hypothesis is that only one of the activators binds the promoter at any given time, and therefore when both are present the chances to bind increases (basically doubling the ON to OFF ratio over time). Moreover, the redundant targets might have longer mRNA half-lives which will contribute to their increase. A more thorough analysis that can help distinguish between these and other potential explanations is needed.

We would like to first clarify the key message and the rationale for the data in **Fig. 3b** and then address the comments by the reviewer.

As we stated in the previous response, the schematic in **Fig. 3a** depicts a hypothesis that cells could modulate the Msn2/4 pulse relationship to regulate gene expression of redundant targets. This hypothesis is based on the findings that the Msn2/4 pulse relationship is modulated in response to different stress conditions (**Fig. 1c**), and pulses of Msn2 and Msn4 can activate target genes in a redundant manner (**Fig. 2b-c**). In this hypothesis, because pulse coincidence rate decreases as stress increases (**Fig. 1c**), pulse coincidence modulation could thus represent a new way for regulating gene expression as stress level changes (**Fig. 3a**). Thus, in **Fig. 3b** we chose a set of stress conditions characterized in **Fig. 1c** (i.e., glucose limitation stresses), and tested the above hypothesis by quantifying the gene expression fold change across four stress levels. Intriguingly, we found that redundant genes display a higher dynamic range between low and high stress compared to non-redundant genes, highly consistent with our hypothesis that pulse coincidence modulation could be used to regulate gene expression. Intuitively, for redundant genes, cells appear to ‘separate’ the coincident Msn2/Msn4 pulses to enhance target

gene activation when stress increases. Such pulse coincidence modulation adds to the modulation in pulse frequency (which is the same for redundant and non-redundant genes), resulting in a higher dynamic range for the redundant genes compared to non-redundant genes.

Based on these clarifications, we now address the concerns raised by the reviewer.

- 1) Regarding the comment that “more TFs driving transcription leads to a higher dynamic range”, we think this is not the main message delivered in **Fig. 3b**. Instead, we emphasized that redundant genes can take advantage of the pulse coincidence modulation to yield higher dynamic ranges compared to non-redundant genes (which could also be regulated by two TFs). In the revised text, we have now clarified the rationales for the experiments in **Fig. 3b (Line 200-224)**.
- 2) Regarding the comment that “accumulated transcript measured in 3b and S4c”, we reason that while it is not as time-resolved as the data in **Fig. 2**, the expression level measured in **Fig. 3b** is a reasonable proxy of the steady-state transcriptional activity. This is because that the transcript half-life is relatively short and thus the effect of transcript accumulation is limited. In the revised text, we have now clarified this point (**Line 215-218**).
- 3) Regarding the alternative hypothesis that “only one of the activators binds the promoter at any given time, and therefore when both are present the chances to bind increases”, we reason that this scenario is not consistent with the data showing how target genes are activated by Msn2/4 pulses. More specifically, as illustrated in the earlier response, each Msn2 or Msn4 pulse can activate redundant target genes near the saturation region of their dose response curves, and the activation level is comparable to a coincident Msn2/4 pulse. Thus, the presence of the second activator, in addition to the first one, does not appear to increase the target activation capacity (i.e., the two activators are functionally redundant), contrary to the picture provided by the alternative hypothesis. To avoid potential confusion, we have now further clarified our hypothesis in the revised manuscript (**Line 200-209, 224-225**).
- 4) Regarding the other alternative hypothesis that “the redundant targets might have longer mRNA half-lives which will contribute to their increase”, we reason that mRNA half-life could affect the absolute expression level, but in theory should not contribute to the relative change in expression level. And in our data, we focused on fold change, a measure of relative change in expression level. To quantitatively explore the relationship between fold change and mRNA half-life, we used mRNA half-life data from two different literatures and did not find correlation between fold change and mRNA half-life (**Fig. R4**). We have now included a note on this point in the revised manuscript (**Line 226-228**).

Figure R4. No correlation is found between mRNA fold change between 0.01% and 0.5% glucose (data from **Fig. 3b**) and mRNA half-life (data from two literatures^{5, 6}).

We believe these changes have addressed the concerns raised by the reviewer and have much improved the manuscript.

3. It is also worth noting that the response to glucose does not seem to saturate under these conditions for either the redundant or non-redundant genes. In order to prove the importance of modulation of coincidence, the comparison should be done under a dose for which the activity is saturated for each TFs individually. This would assure that increase in transcription can only be achieved by modulation of the coincidence rate (or at least eliminate one of the competing hypotheses).

We would like to first further clarify the glucose titration assays and then address the concerns raised by the reviewer.

First, it is important to emphasize that we examined the target responses at the level of individual Msn2/4 pulses. And because the level of glucose limitation stress modulates not only the pulse numbers of Msn2/4, but also the pulse coincidence rate, we thus resorted to glucose limitation stress to access how the modulation of pulse dynamics could be translated into the modulation of target gene expression across different stress levels. A key premise for this assay is that individual pulses of Msn2 or Msn4 can saturate the dose response curves of target genes, which has been established experimentally (**Supplementary Fig. 4d**). Furthermore, we have shown that Msn2 and Msn4 are activated in pulses at the single-cell level in all glucose levels examined, and thus a snapshot of the cell population (by RNA-seq) would recapitulate the time averages of single cells (i.e., the system is ergodic).

Regarding the concern that “the response to glucose does not seem to saturate under these conditions”, it appears that the reviewer was referring to the fact that genes have not yet reached their maximum activation capacities at the cell population level even under the highest glucose limitation stress. We agree that the activation capacities of target genes at the population level have not been saturated. However, the saturation hypothesis in our model refers to the saturation of target responses at the level of individual pulses and at the single-cell level, instead of at the level of the entire cell population. More specifically, at each glucose limitation stress, the

population-level gene activation capacity should relate to the product between the number of effective Msn2/4 pulses and the gene's response to each pulse, where the latter term is saturated with respect to individual Msn2/4 pulses. Therefore, in our model, the gene activation capacity would reach a maximal level when the number of effective pulses maximizes. That said, the fact that genes have not yet reached their maximal activation capacity at the population level does not violate our model hypothesis. To ensure this point is clearly communicated, we have now included more explanations for the rationales of our glucose titration assay (**Line 200-224**).

Regarding the suggestion that “the comparison should be done under a dose for which the activity is saturated for each TFs individually”, as we have stated in the earlier responses, individual pulses of Msn2/4 in single cells have indeed saturated the target genes' transcriptional responses. In the revised text, this point has been clearly mentioned (**Line 172-178, 224-225**).

We believe these changes have addressed the concerns raised by the reviewer and have much improved the clarification of the paper.

4. The authors should display error bars as standard deviation and not standard error. The standard error would have applied if the error bars displayed biological replicates (how likely is the mean to fall within a certain range) but when looking at the distribution of fold changes between many different genes the standard deviation (showing the distribution of fold changes) is more suitable.

We thank the reviewer for the very helpful suggestion. It is indeed not appropriate to use standard error to represent data points from different genes instead of different replicates. To address this, we have now revised the original plots (**Fig. 3b** and original **Extended Data Fig. 4c**) to violin plots showing the distribution of data points. Additionally, we have included p values from Kolmogorov-Smirnov tests.

Fig. 3d, S5 The authors show that the coincidence rate negatively correlates with survivability. However according to the model every pulse should act as an activating pulse. A helpful way to display the data in 3d and S5c is comparing cells by their "effective pulses". That is, in the allotted time (7 hours?) how many unique pulses did a cell experience (so a coinciding pulse would be counted as one). While this can be derived from the heatmap in figure 3d and S5c, this type of presentation would be easier to understand and will allow direct comparisons between the WT and the mutants (extended 5d-e).

We thank the reviewer for the very helpful suggestion. Based on our model prediction, cells with a higher number of effective pulses should possess a higher stress survival capacity. The new plot shows that indeed, as the number of effective pulses increases, the stress survival capacity increases accordingly in the wild-type cells. This positive correlation is much stronger compared to the mutant cells where the DBDs of Msn2/4 have been deleted (**Fig. 3e** and

Supplementary Fig. 5e).

We note that there appears to be some correlation between pulse number and survival in the DBD mutant (which was also present in the original heatmap representation of the data), which is likely due to Msn2/4-independent regulation (or leftover Msn2/4-dependent regulation, see responses to a latter question). Such Msn2/4-independent regulation might likely account for the overall enhanced survival capability in the DBD mutant cells. It is important to note that these observations do not contradict with our model. This is because that our model is compatible with the presence of survival regulations other than Msn2/4, and most importantly, the key prediction of our model is validated by the data showing that cells with same Msn2/4 pulse numbers have differential survival capacities depending on the pulse coincidence rate. In the revised manuscript, we have now clearly discussed these points and noted the presence of Msn2/4-independent regulation of cell survival (**Line 301-304**).

In order to further investigate our model, we incorporated the reviewer's suggestion and tested another model prediction that, with the same number of effective versus total Msn2/4 pulses, cells should possess different survival capabilities. This is because that there are coincident Msn2/4 pulses, and Msn2/Msn4 pulses are functionally redundant. More specifically, our model predicts that cells with a certain number of effective pulses would have higher survival capabilities compared to cells with the same number of total pulses. To test this, we calculated the stress survival fractions for groups of cells with different effective pulse numbers and with corresponding total pulse numbers. We found that with the same number of total pulse or effective pulse, the latter group of cells has a higher stress survival capacity for almost all the pulse numbers analyzed. This result offers a key line of evidence supporting our model (see **Fig. 3e, Line 277-286** in the text).

Furthermore, analogous to the above prediction, our model further predicts that adding an extra effective pulse would yield a larger increase in cell survival capability compared to adding an extra total pulse. This is because an extra total pulse could coincide with existing pulses (and thus would not help the cell), while an extra effective pulse (by definition) would lead to additional expression of stress response genes. To test this, we computed the fold-change between the cell survival contributions by a single effective pulse and by a single total pulse. We found that under most scenarios, adding an extra effective pulse indeed results in a larger contribution to cell survival compared to adding an extra total pulse. This result further substantiates our model (see **Supplementary Fig. 5c, Line 288-294** in the text).

Together, we believe that these changes have allowed direct comparisons between wild type and other mutants. And the new results have provided further supports for our model that yeast cells can modulate pulse coincidence rate to regulate stress survival.

Furthermore, when looking at 3d I'm a bit lost as to why, if the model is correct, cells with 5 pulses in which 2 coincide, have lower survival than cells with 3 pulses with 0 coincidence (same number of unique pulses). Theoretically they have more chances to activate transcription

even though they only effectively experience 3 pulses (according to the model). The way the data is displayed now makes it look as if the relationship holds only when looking at cells with equal frequencies and not across frequencies, and if so it should be discussed.

We thank the reviewer for raising an issue that requires clarification. What the reviewer stated is accurate -- we initially intended to only compare among cells that have the same total number of Msn2 plus Msn4 pulses. This was somewhat unfortunate because we did not have enough cells in the original dataset (as it was a non-trivial task to collect long time trajectories for a large number of cells). To address this limitation, we have carried out additional experiments to obtain a much larger dataset, which allowed us to perform more quantitative analysis for the dependence of cell survival on pulse coincidence rate as well as on pulse frequency. The details of the analysis are presented in the response to the next concern, which should address the issue raised here.

Lastly, the analysis in 3d and 5c is limited due to the cutoff at 6 pulses. The small number of cells in each condition can be solved by binning cells into groups which will also strengthen the statistics (look at cells with 1-2 pulse, 3-5 pulses, 6-8 etc.). If there is a specific reason for the cutoff it should be explained.

We thank the reviewer for raising a limitation in our original dataset. We chose the cutoff at 6 pulses since the cell numbers in higher pulse number bins were small. We address this issue by acquiring much more cells and by implementing new analyses.

To address the issue, we carried out new additional survival assay experiment, and have now obtained more than 1000 cells that were monitored over 13 hours of movie at 2-minute frame interval. With this new dataset, we performed several complementary analyses to test our model and model predictions.

- 1) In the first analysis, we tested that for cells with the same total Msn2/4 pulse number, whether the ones with low pulse coincidence rates would have a higher stress survival capability. This analysis was originally carried out in **Fig. 3d**, but because each subset contained a relatively small number of cells and it was challenging to arrive at a statistically significant conclusion. To enhance the statistical power, we binned cells having similar pulse numbers (e.g., 1-2, 3-4, 5-6 pulses), and obtained two sub-groups of cells for each binned group of cells based on the pulse coincidence rate in each cell (see **Methods** for details). By doing so, we increased the number of cells for each sub-group and thus enhancing statistical power, and we could also test our model for cells with more pulses (larger than 6 pulses). We found that for six different groups of cells (containing varying pulse numbers), the sub-groups with lower pulse coincidence rates all have higher stress survival rates compared to their counterparts. Additionally, cells with more total pulse numbers have higher stress survival rates. And importantly, these relationships are abolished in the DBD mutant strain. These results provide a strong support for our model (see **Fig. 3d** and **Supplementary Fig. 5d, Line 256-264, 296-301** in the text).
- 2) In the second analysis, we aimed to test our model in a more quantitative manner by

constructing linear regression models, which was originally carried out in the original **Extended Data Fig. 5b**. With the new updated dataset, we binned cells based on the total pulse number and the number of coincident pulses, and repeated the analysis. We found that the linear model including the number of coincident pulses can better explain the survival capabilities of different cell groups, providing an additional support for our model (see **Supplementary Fig. 5b, Line 266-275** in the text).

Additionally, we included new analyses showing that cell survival also depends on the number of the effective pulse, which have been discussed in the earlier response. Together, these data show that the stress survival capacity of yeast cell could be modulated by either the effective pulse number or the pulse coincidence rate, which have now much strengthened our main conclusion.

S5c The authors claim that the correlation between coincidence and survival is dependent on transcription, as mutants of Msn2 and 4 that are unable to bind DNA don't show this correlation. However, based on Figure S5c it seems that cells lacking Msn2 and Msn4 survive better (50% survival vs 20% with the WT genes) and that the correlation is inverted (more coincidence leading to more survival). I might have misunderstood this analysis, but at the moment I find it confusing and contradictory to the main conclusions of the paper.

We thank the reviewer for raising questions regarding the survival data with the DBD mutant strain. We would like to first address the trend in the original plot with the DBD mutant cells, where cells with more coincident pulses appeared to survive longer. We have determined that this apparent correlation was likely due to the relatively low cell number. This is because that in the original analysis, the cell numbers in the bins with more coincident pulses are relatively small (10-20 cells). In the new analysis suggested by the reviewer, we binned cells with similar pulse numbers. Additionally, we separated cells with similar pulse numbers into only two subgroups that have either low or high pulse coincidence rate. By doing so, we have more cells in each subgroup. With this new analysis, we found that for the DBD mutant cells, there is no apparent correlation between survival fraction and pulse coincidence rate (see **Supplementary Fig. 5d**).

For the second observation that the strain with the deletion of Msn2DBD and Msn4DBD exhibited an apparent increase in stress survival. This is indeed puzzling and interesting. However, because this phenomenon could be related to the genetic perturbations caused by DBD mutation, it does not contradict with our main conclusions, which were based on experiments with the wild-type cells. More specifically, the increased stress survival capability in the mutant strain could result from mechanisms other than Msn2/4. For example, it is reasonable to assume that there are additional mechanisms contributing to stress survival such as other PKA-regulated TFs. And when Msn2 and Msn4 are mutated, these mechanisms would dominate the control of stress survival in the mutant strain (and surprisingly contribute to better survival). It is noted that we did not completely delete Msn2/4 but only deleted the DBD domains, which could cause unknown perturbations to stress response.

Importantly, this potential explanation does not contradict with our model conclusion. This is because that when comparing the stress survival for cells with the same total pulse number, only the wild-type cells display a consistent dependence on pulse coincidence rate for all pulse numbers, while the DBD mutant does not display a consistent pattern. Such a distinction is much more obvious in our updated analysis compared to our original analysis (**Fig. 3d** and **Supplementary Fig. 5d**). In the revised manuscript, we have included a discussion on this issue to aid interpreting the interesting feature in the DBD mutant data (**Line 301-304**).

Fig S5d and e The authors show that in cells expressing either Msn2 or Msn4 increased pulse frequency correlates to increased survival. Other models should be considered here, including integrated Msn level, or duration of Msn level above a certain threshold as determinant of survival. In other words, does it matter whether low frequency is achieved by having sharp, narrow peaks of activity with long periods of 0 activity, vs. having wide broad peaks of activity with minimal time at 0 activity?

We thank the reviewer for these suggestions. We have performed additional analyses to investigate how stress survival capacities in both *msn2* and *msn4* strains could be explained by the suggested features. As shown in **Supplementary Fig. 5f-h**, integrated Msn2/4 levels (namely, pulse frequency multiplied by pulse area) were not as significant as Msn2/4 frequency alone to explain survival ability. This is consistent with our model that pulse area (or pulse duration/amplitude) does not contribute to higher survival ability compared to pulse number. Mechanistically, this observation could be explained by what we proposed in the original manuscript, i.e., individual pulses of Msn2/4 ‘hit’ the saturation region of the target gene’s dose response curve, which has been further validated experimentally in our new figure panel (**Supplementary Fig. 4d**).

Fig. S6 The results presented in this figure are problematic. The assay does not prove that Msn4 has higher affinity for PKA, but only that Msn4 is less active when driven under the same promoter as Msn2. It is unknown, for example, if the levels of these two constructs under these conditions is similar. This needs to be shown experimentally by western blots or imaging. In addition, since the affinity argument is central to the model, an in vitro binding assay is needed and an in vitro kinase assay would be helpful too.

We thank the reviewer for these helpful suggestions. The reviewer raised two important concerns: a) whether the difference in nuclear localization levels could be explained by the difference in protein levels; b) whether there are additional evidences supporting the difference in the affinities to PKA.

To address the first concern, we carried out two different assays to test whether the difference in nuclear localization levels could be explained by the difference in protein levels. More specifically, we aimed to test the hypothesis that for the two mutant strains shown in

Supplementary Fig. 6a, the reduced nuclear localization level of Msn4 arises from the reduced Msn4 protein expression level compared to Msn2, rather than from Msn4's higher affinity to PKA. A key premise of this hypothesis is that the protein expression level of Msn4 is lower than that of Msn2.

- 1) To test this, we quantified cellular Msn2 and Msn4 protein levels in these two mutant strains using both flow cytometry and fluorescence microscopy, as both proteins are fused with CFP. Flow cytometry data showed that the CFP signal distribution for Msn4-CFP is slightly right shifted compared to that for Msn2-CFP, indicating that there are more Msn4 protein molecules (see **Supplementary Fig. 6c, Line 357-358** in the text). In the second assay, cellular CFP quantifications using fluorescence microscopy showed that mean Msn4-CFP level per cell is ~7.4% higher than Msn2-CFP (see **Supplementary Fig. 6d, Line 357-358** in the text).
- 2) Results from both techniques show that the mean Msn4 protein level is slightly higher than Msn2, which likely could not explain the observed differences in their nuclear localization levels (i.e., $Msn4 < Msn2$). In other words, the differential localization levels are likely due to the differential affinities to PKA, as we originally hypothesized.

To address the second concern, we aimed to collect more data to support our hypothesis that the different nuclear localization levels are due to the differential affinities to PKA. In the original manuscript, we only had one piece of evidence supporting this hypothesis. To address this, we carried out three different assays.

- 1) In the first assay, we aimed to compare the nuclear localization dynamics of Msn2 and Msn4 in response to sudden inhibition of PKA activity. Our model predicts that if Msn2/4 possess differential affinities to PKA, then their nuclear localization responses to PKA inhibition would be different. More specifically, the TF with a lower affinity to PKA would be dephosphorylated faster, resulting in a more rapid shuttling into the nucleus. Using the two mutant strains illustrated in **Supplementary Fig. 6a**, we measured the nuclear localization responses of initially cytoplasm localized Msn2-CFP and Msn4-CFP upon the addition of H-89, a validated PKA inhibitor in budding yeast. The data showed that the PKA inhibitor H-89 induces a much more rapid cytoplasm-to-nucleus shuttling of Msn2-CFP compared to Msn4-CFP. In other words, this data suggests that under the same inhibited PKA activity, Msn2 is much more easily dephosphorylated (and shuttled into the nucleus) compared to Msn4, providing a strong support for our model that Msn2 has a lower affinity to PKA than Msn4 (see **Supplementary Fig. 6b, Line 347-356** in the text).
- 2) In the second assay, we further compared the nuclear localization dynamics of Msn2-CFP and Msn4-CFP under glucose limitation stress. During glucose limitation, PKA is first inhibited and then re-activated. When examining the nuclear localization responses of Msn2/4, we observed a clear difference in the dynamics of the response, i.e., while both TFs enter the nucleus with the same timescale, Msn4 displays a much faster timescale of exit from the nucleus compared to Msn2. This data suggests that Msn4 is re-phosphorylated by PKA at a faster time scale, which is consistent with the picture that Msn4 has a higher affinity to PKA compared to Msn2 (see **Supplementary Fig. 6g, Line 358-364** in the text).

- 3) In the third assay, we further compared their nuclear localization dynamics under both KCl and ethanol stresses. Both stresses induce fast nuclear localizations of Msn2 and Msn4, but the transient pulses of Msn4-CFP had a significantly lower amplitude compared to that of Msn2-CFP for both stresses. This result is consistent with the picture that Msn4 is more phosphorylated by PKA and thus its nuclear localization is weaker compared to Msn2. This data thus further substantiates our hypothesis (see **Supplementary Fig. 6e-f, Line 358-364** in the text).
- 4) New evidences from these three assays together provide strong supports for our model that Msn2 and Msn4 have differential affinities to PKA and their nuclear localization dynamics are differentially regulated by PKA.

Together, we believe that these results further substantiate our model and have greatly strengthened our manuscript.

Fig 4 and S7 The authors propose factor X as an inhibitor of Msn4, which already has higher affinity to PKA. There are other alternative models that can explain the data and should be discussed. For example, the increased affinity of PKA to Msn4 might be dependent on X. In this model, a decrease in X will reduce the affinity of PKA to Msn4 below the affinity to Msn2, allowing for an Msn4 pulse in ranges that would still be inhibitory to Msn2.

We thank the reviewer for raising an important issue regarding alternative models for pulse coincidence modulation. We apologize for not providing related information in the original manuscript. Based on the comments by both reviewer #2 and reviewer #3, we have now compiled a list of alternative models that could enable pulse coincidence modulation as well as a list of four experimental observations that the model should be consistent with (see **Supplementary Fig. 8a**). For each potential model, we determined whether each of the experimental observations is consistent with the model. By doing so, we found that the model we proposed originally is so far the only one that could be consistent with all four experimental observations. However, in the revised manuscript, we emphasized that it is likely that we have missed some important mechanisms, and our model is only meant to offer explanations for the current experimental observations (**Line 408-414**). Thus, while our model may not be the only model that explains our single-cell data, it sheds a light into how the divergent temporal dynamics of paralogous TFs could be achieved.

Regarding the specific model raised by the reviewer, it is very interesting and closely resembles our proposed model. However, this specific model could not explain the phenomenon that Msn4 instead of Msn2 still pulses in the absence of PKA⁷, which was why we assumed that X regulated Msn4 directly instead of through changing the affinity of PKA to Msn4. For details of this and other alternative models, please refer to **Supplementary Fig. 8a**.

To make the model meaningful the authors should clarify whether the parameters were derived from empirical measurements or were picked for best fit. For example, are the Msn2/ 4

activation thresholds based on real data? The figure legend states Msn2 and Msn4 binding affinities in a ratio of 0.5—what data supports that? A sensitivity/robustness analysis is needed for the chosen parameters.

We would like to first re-emphasize that our model is a phenomenological model instead of a mechanistic model (which was explicitly stated in our original manuscript, see “Mathematical model description” section in **Methods**). Because of the phenomenological nature of the model, many details of the model were thus simplified or omitted. And the model parameters were chosen based on empirical values that are necessary to qualitatively satisfy the key observations in our experiments. For example, our experiments showed that PKA has a higher affinity for Msn4 than for Msn2, and thus in our model, the ratio between the dissociation constants for PKA regulating Msn4 and Msn2 was chosen from 0.2 to 1. The Msn2/4 activation thresholds were thus related to the above dissociation constants. In the original **Extended Data Fig. 7b**, we simply chose a value between 0.2 and 1 (namely, 0.5) to show an example single-cell trajectory.

To address the reviewer’s concern regarding sensitivity/robustness analysis, we have now performed new analyses of the model using a range of key parameters. More specifically, in our model, the key parameters were PKA level and the ratio of dissociation constants for PKA regulating Msn4 and Msn2. As shown in **Supplementary Fig. 8f-g**, under a wide range of parameter values, we consistently observed the phenomenon that increasing PKA or decreasing PKA affinity difference (namely, increasing the ratio of dissociation constant for PKA regulating Msn4 and Msn2) increased pulse coincidence rate and changed the distribution of pulse coincidence rate. These results indicate the robustness of the model behaviors to the chosen parameters.

Fig S7h In order to show that the coincidence rate drops in the Msn2 mutants, the authors need to demonstrate that the pulsatile dynamics of Msn2 are unaltered. One prediction could be that the coincidence drops because these mutants are inactive, or do not exhibit a pulsatile dynamic anymore. Without these additional controls it is difficult to support the authors’ claims.

We thank the reviewer for these comments. We think that the reviewer might have missed our results in the original **Extended Data Fig. 7f**, which showed that the mutants were not inactive but instead they showed even higher Msn2 pulse frequency since PKA showed lower affinity for Msn2 in these strains than in wild type. Thus, the drop of coincidence rate was not because that these mutants were inactive. We have included a sentence in the revised text to clarify this (**Line 403-404**).

Page 11 “Thus, ancestral Msn2/Msn4 acquired pulsing before WGD, consistent with our hypothesis that a mechanism for modulating their temporal relationship was evolved post duplication to preserve their functional redundancy.”

Since *K. lactis* and *S. pombe* have only one Msn gene (which pulses in *K. lactis*), it is unclear how this proves that a mechanism for modulating Msn2/4 temporal relationship evolved post-genome duplication. It couldn't evolve preduplication, since there was only one Msn gene so there was no "relationship" to modulate. The authors have only proven that pulsing evolved pre-WGD. Moreover, the authors don't know WHY it evolved, they just claim that it supports fitness thus contributing to preservation. This statement needs to be adjusted.

We thank the reviewer for these comments. It is true that temporal redundancy modulation did not evolve pre-WGD since there was only one Msn2/4 ortholog and thus no relationship. To clarify our model, we think that after WGD, the two paralogs appeared and the relationship also appeared, which then could undergo the evolutionary process as we proposed. Mechanically, divergence of temporal dynamics between two paralogs increased potentially due to PKA's differential affinity (perhaps due to mutations in DNA sequences around PKA sites) and the involvement of other kinase regulations, and this together with functional redundancy could enhance stress survival ability. Thus, this fitness could contribute to the evolutionary preservation of functional redundancy.

Regarding the reason why pre-WGD Msn needed to evolve temporal dynamics, as pointed out by the reviewer, we indeed have no evidence supporting the necessity of pulsing in the evolutionary timescale. Nevertheless, in order for post-WGD Msn pair to preserve functional redundancy, the pair of paralogs would need to evolve divergence at the level of cis-regulation, protein function, or temporal dynamics (as we demonstrated). And indeed, as the reviewer pointed out, we cannot rule out divergence at other levels that might also promote the preservation of functional redundancy. In the revised manuscript, we have made sure to emphasize the limitations of our model for explaining the preservation of functional redundancy (**Line 526-531**). Additionally, since functional redundancy and dynamics divergence did not exist before WGD, we also modified **Fig. 5e**.

Minor comments

Introduction:

"While the transient response dynamics can encode information about the inputs, the sustained dynamics after the transient phase enable complex signal processing mechanisms". There is very little explanation of the different phases of the dynamical response in the introduction. A more detailed explanation is needed especially for less systems-oriented readers.

We thank the reviewer for the helpful suggestion. Indeed, we should have included more explanations on this. We have added more detailed explanations about transient phase and steady-state phase in the revised manuscript (**Line 58-62**).

Page 6: "It is interesting to note that, by altering temporal relationship between functionally redundant Msn2 and Msn4 pulses, cells can dynamically modulate the degree of redundancy

between Msn2 and Msn4, resulting in an effective modulation of redundancy in a temporal manner (i.e., temporal redundancy modulation).” At this point of the paper this statement should be presented as an hypothesis.

We thank the reviewer for pointing out this issue and we admit that this statement should be a hypothesis if positioned before the data. We have changed this statement to a hypothesis and further showed our data as its proof in the revised manuscript (**Line 200-209**).

“We next analyzed how cells could benefit from the modulation of temporal redundancy.” This should also be changed since modulation of redundancy has not been demonstrated.

We have clarified the terminology of “temporal redundancy modulation” in the revised text, and made sure that it is appropriately located (**Line 230-235**).

Page 9 “We identified a candidate kinase Yak1” It will be helpful to elaborate how this kinase was identified – was it through a screen? Literature search? An educated guess? Were there are other potential candidates?

We thank the reviewer for pointing out this issue. We apologize for not describing how we found Yak1 as kinase X. We have now included more details regarding the screen process (**Line 371-381**). First, we did literature research about potential regulators of Msn2 or Msn4, including CKA2, FPK1, PTK2, RIM15, SNF1 and YAK1. Then we constructed copper-inducible strains for tuning the expression levels of these regulators and conducted microscopies of these strains under different copper levels (namely, different regulator levels) to compute the change of Msn2 pulse frequency and Msn4 pulse frequency. Finally, we found that when elevating Yak1 level, Msn4 frequency decreased while Msn2 frequency did not change, which indicated that Yak1 may be a regulator inhibiting Msn4 alone (the schematic of this pipeline has been added as **Supplementary Fig. 7b**).

To strengthen our hypothesis that Yak1 is kinase X which represses Msn4 alone, we conducted a new experiment (containing many more cells than the original dataset) to compare Msn2/4 pulse frequency under different Yak1 levels. Briefly, we cultured the strain with copper-inducible CUP1 promoter for perturbing Yak1 level with different levels of Cu²⁺ and took images for bright-field, Msn2 (CFP) and Msn4 (YFP). Then single cells were segmented and nuclear localization of Msn2 and Msn4 was identified for each cell (see **Methods** for more details). As shown in **Supplementary Fig. 7e**, elevating Yak1 level only decreased the percentage of cells with nuclear localization of Msn4 while did not change that of Msn2, indicating that increasing Yak1 decreased Msn4 pulse frequency alone. This result, together with **Supplementary Fig. 7d**, further supports that Yak1 appears to be the kinase X repressing Msn4 alone in our experimental system. However, our screen is relatively limited and we have now clearly stated this limitation and have discussed the possibility of other kinases or phosphatases regulating the dynamics of Msn4 (**Line 408-410**).

1. N. Hao, E. K. O'Shea, Signal-dependent dynamics of transcription factor translocation controls gene expression. *Nat Struct Mol Biol* **19**, 31-39 (2011).
2. A. S. Hansen, E. K. O'Shea, Promoter decoding of transcription factor dynamics involves a trade-off between noise and control of gene expression. *Mol Syst Biol* **9**, 704 (2013).
3. R. Martinez-Corral, E. Raimundez, Y. Lin, M. B. Elowitz, J. Garcia-Ojalvo, Self-Amplifying Pulsatile Protein Dynamics without Positive Feedback. *Cell Syst* **7**, 453-462 e451 (2018).
4. D. Chen *et al.*, Multiple pathways differentially regulate global oxidative stress responses in fission yeast. *Mol Biol Cell* **19**, 308-317 (2008).
5. C. Miller *et al.*, Dynamic transcriptome analysis measures rates of mRNA synthesis and decay in yeast. *Mol Syst Biol* **7**, 458 (2011).
6. S. E. Munchel, R. K. Shultzaberger, N. Takizawa, K. Weis, Dynamic profiling of mRNA turnover reveals gene-specific and system-wide regulation of mRNA decay. *Mol Biol Cell* **22**, 2787-2795 (2011).
7. M. Jacquet, G. Renault, S. Lallet, J. De Mey, A. Goldbeter, Oscillatory nucleocytoplasmic shuttling of the general stress response transcriptional activators Msn2 and Msn4 in *Saccharomyces cerevisiae*. *J Cell Biol* **161**, 497-505 (2003).

REVIEWERS' COMMENTS

Reviewer #1 (Remarks to the Author):

The authors have addressed my concerns satisfactorily. Therefore, I recommend the acceptance of the manuscript for publication.

Nan Hao

Reviewer #2 (Remarks to the Author):

I am satisfied by the authors' response to my comments. I am extremely impressed by the amount of work that was performed to address the issues raised, especially revisions made to Fig. 3 and Supp. Figs 5 and 7. I also appreciate the efforts made by the authors to better support their hypothesis that Yak1 is kinase X.

Overall, the authors were able to address all issues raised in my report. The changes made to the manuscript greatly strengthen it and I now find it suitable for publication.

Reviewer #3 (Remarks to the Author):

The authors have satisfactorily addressed my major concerns and comments, and I support publication of the revised study in Nature Communications. I have 2 minor comments, which I encourage the authors and editor to consider:

First, Figure 2B indeed shows redundancy in the synthetic gene. However, the DDR2 increase in the Msn2/4 double doesn't seem statistically significant. It will be helpful to clarify that and if indeed this is the case include a short acknowledgement of that.

In addition, in response to comment #2, the authors explained that their steady state RNA measurement can be used as a proxy for transcription rate because mRNA half-lives are short. "Short" is not a quantitative term, and is especially difficult to understand without mentioning in comparison to what. A more accurate statement would be that mRNA half-lives are short compared to the steady state measurement (around 16h).

Point-by-point reply (reviewer comment in blue and author response in black):

Reviewer #1 (Remarks to the Author):

The authors have addressed my concerns satisfactorily. Therefore, I recommend the acceptance of the manuscript for publication.

Nan Hao

We are gratified by the positive comments. We greatly appreciate all the helpful comments and suggestions by the reviewer, which have significantly improved our manuscript.

Reviewer #2 (Remarks to the Author):

I am satisfied by the authors' response to my comments. I am extremely impressed by the amount of work that was performed to address the issues raised, especially revisions made to Fig. 3 and Supp. Figs 5 and 7. I also appreciate the efforts made by the authors to better support their hypothesis that Yak1 is kinase X.

Overall, the authors were able to address all issues raised in my report. The changes made to the manuscript greatly strengthen it and I now find it suitable for publication.

We are gratified by the positive comments. We greatly appreciate all the helpful comments and suggestions by the reviewer, which have significantly improved our manuscript.

Reviewer #3 (Remarks to the Author):

The authors have satisfactorily addressed my major concerns and comments, and I support publication of the revised study in Nature Communications. I have 2 minor comments, which I encourage the authors and editor to consider:

We are gratified by the positive comments. We greatly appreciate all the helpful comments and suggestions by the reviewer, which have significantly improved our manuscript.

First, Figure 2B indeed shows redundancy in the synthetic gene. However, the DDR2 increase in the Msn2/4 double doesn't seem statistically significant. It will be helpful to clarify that and if indeed this is the case include a short acknowledgement of that.

We thank the reviewer for these helpful comments. We agree that it would be helpful to clarify a noticeable feature in the DDR2 data, where DDR2's response to Msn2/4 co-pulse appears slightly stronger (yet statistically insignificant) compared to Msn2-only or Msn4-only pulse. It is important to point out that this feature does not contradict with our model.

We have now included a sentence in the main text to clarify this point (**Line 153-154**), which reads: “Note that *DDR2* exhibits a slightly stronger (but statistically insignificant) response under *Msn2/4* co-pulse compared to that under *Msn4*-only or *Msn2*-only pulses.”

In addition, in response to comment #2, the authors explained that their steady state RNA measurement can be used as a proxy for transcription rate because mRNA half-lives are short. “Short” is not a quantitative term, and is especially difficult to understand without mentioning in comparison to what. A more accurate statement would be that mRNA half-lives are short compared to the steady state measurement (around 16h).

We thank the reviewer for these helpful comments. We agreed that the rationale underlying the steady-state RNA-seq assay design should be further clarified. We have now added a clarification of the assay in the main text (**Line 219-220**), which reads: “The rationale is that mRNA half-lives are short compared to the steady state measurement (~ 4 hours)...”